# Targeting endogenous kidney regeneration using anti-IL11 therapy in acute and chronic models of kidney disease

Anissa A. Widjaja [1,8] ✉, Sivakumar Viswanathan [1], Shamini G. Shekeran[1], Eleonora Adami[1,2], Wei-Wen Lim [1,3], Sonia Chothani[1], Jessie Tan[3], Joyce Wei Ting Goh [1], Hui Mei Chen [1], Sze Yun Lim[1], Carine M. Boustany-Kari[4], Julie Hawkins[4], Enrico Petretto [1], Norbert Hübner [2,5,6], Sebastian Schafer [1], Thomas M. Coffman[1] & Stuart A. Cook [1,3,7,8] ✉

The kidney has large regenerative capacity, but this is compromised when kidney damage is excessive and renal tubular epithelial cells (TECs) undergo SNAI1-driven growth arrest. Here we investigate the role of IL11 in TECs, kidney injury and renal repair. IL11 stimulation of TECs induces ERK- and p90RSK-mediated GSK3β inactivation, SNAI1 upregulation and pro-inflammatory gene expression. Mice with acute kidney injury upregulate IL11 in TECs leading to SNAI1 expression and kidney dysfunction, which is not seen in *Il11* deleted mice or in mice administered a neutralizing IL11 antibody in either preemptive or treatment modes. In acute kidney injury, anti-TGFβ reduces renal fibrosis but exacerbates inflammation and tubule damage whereas anti-IL11 reduces all pathologies. Mice with TEC-specific deletion of *Il11ra1* have reduced pathogenic signaling and are protected from renal injury-induced inflammation, fibrosis, and failure. In a model of chronic kidney disease, anti-IL11 therapy promotes TEC proliferation and parenchymal regeneration, reverses fibroinflammation and restores renal mass and function. These data highlight IL11-induced mesenchymal transition of injured TECs as an important renal pathology and suggest IL11 as a therapeutic target for restoring stalled endogenous regeneration in the diseased kidney.

It is estimated that 1 in 4 adults over the age of 60 suffer from chronic kidney disease (CKD) that is characterized by loss of kidney function and fibrosis, regardless of primary etiology. Recovery from established CKD is unusual, but in rare documented cases, reversal of structural lesions of the glomerulus and tubulointerstitium have been described[1–3].

Acute kidney injury (AKI) is caused by injury to proximal tubular epithelial cells (TECs) from causes such as ischemia or toxins and is typically reversible because of the substantial regenerative capacity of TECs[4]. When the extent of kidney injury exceeds its intrinsic regenerative capacity, such as after severe or repeated episodes of AKI,

[1]Cardiovascular and Metabolic Disorders Program, Duke-National University of Singapore Medical School, Singapore, Singapore. [2]Cardiovascular and Metabolic Sciences, Max Delbrück Center for Molecular Medicine in the Helmholtz Association (MDC), 13125 Berlin, Germany. [3]National Heart Research Institute Singapore, National Heart Centre Singapore, Singapore, Singapore. [4]Boehringer Ingelheim, CardioMetabolic Disease Research, Berlin, Germany. [5]DZHK (German Centre for Cardiovascular Research), Partner Site Berlin, 13347 Berlin, Germany. [6]Charité-Universitätsmedizin, 10117 Berlin, Germany. [7]MRC-London Institute of Medical Sciences, Hammersmith Hospital Campus, London, UK. [8]These authors contributed equally: Anissa A. Widjaja, Stuart A. Cook. ✉e-mail: anissa.widjaja@duke-nus.edu.sg; stuart.cook@duke-nus.edu.sg

fibrosis, and inflammation ensue, leading to chronic structural changes and lasting kidney functional impairment.

Operating at the interface of regeneration and fibrosis is the maladaptive process of epithelial–mesenchymal transition (EMT) of injured TECs, a process regulated by expression of the E-Cadherin repressor *SNAI1*, which is controlled by TGFβ1 and other factors[5–8]. SNAI1 expression is tightly regulated by GSK3β activity, which is itself inhibited by ERK and p90RSK phosphorylation[7,9]. As such, SNAI1 and GSK3β act together as an ERK-regulated molecular switch for EMT[9,10]. The specific EMT process operating in TECs has been termed partial EMT (pEMT), also coined a 'failed-repair proximal tubule cell' state[11], as it is localized to damaged epithelial cells and does not generate interstitial myofibroblasts, which was a point of some contention in the earlier literature[5,6].

TEC plasticity is profound and, in the injured kidney, the balance between TEC proliferation and pEMT determines whether the kidney regenerates or scars[12]. Achieving therapeutic inhibition of pEMT has been a long-term goal in the search for curative treatments for CKD. Unfortunately, antibody neutralization of TGFβ failed in clinical trials, related to on-target pro-inflammatory toxicities, and new approaches for treating CKD are needed[13,14].

We recently identified interleukin 11 (IL11) as a pro-fibrotic factor in fibroblasts and showed that IL11 contributes to renal fibrosis in a mouse model of AKI[15] and renal failure in genetically-driven glomerular disease[16]. IL11 can also stimulate epithelial cell EMT[17,18] and in epithelial cancer cells, TGFβ1 specifically upregulates *SNAI1*, *HAS2*, and *IL11*[19]. Here we hypothesized that IL11, which activates ERK signaling required for EMT[9,10,20], may cause TEC mesenchymal transition and dysfunction as an initiating pathology in kidney disease.

In this work, we show that damaged TECs both produce and respond to IL11 to activate ERK/p90RSK, inactive GSK3β, and initiate a SNAI1-driven program of TEC dedifferentiation. Mice deleted for *Il11* are protected from AKI and have reduced pEMT, inflammation, and fibrosis, as well as better renal function. In mouse models of AKI or accelerated chronic kidney disease, administration of anti-IL11 is associated with lesser renal pathology and improved renal function. While anti-TGFβ increases renal inflammation and tubule damage in AKI, due to on-target toxicities, these side effects are not seen with anti-IL11. Conditional deletion of *Il11ra* in TECs reduces pathogenic signaling and tissue damage and preserves renal function. In a model of CKD, anti-IL11 reverses renal pathology, enables TEC proliferation, and restores kidney parenchymal mass and renal function. These data identify IL11 signaling in TECs as a key pathology for stalled kidney regeneration and reveal IL11 as a potential new therapeutic target for unleashing endogenous repair mechanisms in acute and chronic renal diseases.

## Results

### Autocrine IL11 activity causes tubular epithelial cell mesenchymal transition

To explore the cellular source of IL11 in damaged kidneys we subjected reporter mice with EGFP knocked into the *Il11* locus (*Il11^EGFP/+^*)[21] to folic acid (FA)-induced AKI[15]. Three days after injury, a robust EGFP signal was detected in TECs (Fig. 1a). By day 28 post AKI, EGFP was less common in TECs and mostly seen in interstitial regions (Fig. 1a). Thus, following AKI, IL11 is first detected in the epithelium but, at later time points, more prominently expressed in the interstitium.

To determine whether TECs might also be a target for IL11, we looked for IL11RA by immunofluorescence staining and confirmed IL11RA expression in primary human renal TECs (Fig. 1b), consistent with human protein atlas data using three independent antibodies (HPA013162, HPA036652, CAB032830)[22]. Stimulation of TECs with TGFβ, a potent inducer of pEMT[5,23], strongly induced IL11 secretion, suggesting an autocrine loop of IL11 secretion and activity in TECs undergoing pEMT (Fig. 1c).

IL11 causes short-term STAT3 and sustained ERK activation across multiple cell types[18,24–26]. We assessed IL11-stimulated STAT3 and ERK activation in TECs and determined if this was linked with mesenchymal transition. Similar to other cells, IL11-induced phosphorylation of STAT3 (pSTAT3), although with a prolonged profile as compared to fibroblasts (Fig. 1d)[25]. Unexpectedly, pERK was significantly decreased early on (from 15 min to 4 h), when pSTAT3 was increased. We probed for pMEK, upstream of pERK, that was elevated early on (15 min) and increased further over time (Fig. 1d). At later time points, SNAI1 increased, and E-Cadherin decreased, which was coincident with elevated pERK, pp90RSK, and pGSK3β. As such, IL11 induces mesenchymal transition of TECs that is associated with an IL11-induced ERK activation profile that differs from other cell types.

### IL11 induces pro-inflammatory and mesenchymal genes in tubular epithelial cells

The signaling data suggested that IL11-stimulated, pMEK-induced pERK levels might be suppressed by a phosphatase. To investigate this, and to study IL11-related transcriptional changes in TECs more generally, we performed RNA-sequencing of TECs stimulated with IL11 for 1, 6, or 24 h.

Principal component analysis (PCA) revealed changes in gene expression across the time course (Fig. 1e). Examination of gene expression by molecular signatures database (MSigDB) analysis showed significant increases in a range of gene set hallmarks that included: TGFβ signaling, IL6/JAK/STAT3 signaling, mTORC1 signaling, inflammatory and IFNγ responses, TNFα signaling, epithelial–mesenchymal transition and G2M checkpoint (Supplementary Fig. 1). These data reinforce the importance of IL11 for TGFβ effects[15] and the activation of mTORC1 by IL11[18,26] while suggesting a role for IL11 in TEC mesenchymal transition and inflammation.

We next examined changes in expression of individual genes in an effort to identify a candidate phosphatase that might inactivate pERK (Fig. 1d). Remarkably, the most significantly upregulated transcript (30.8-fold, $P = 2.2 \times 10^{-308}$) genome-wide 1 h post stimulation was *DUSP5* (Fig. 1f; Supplementary Data 1), which is an ERK phosphatase that limits inflammatory responses[27,28]. DUSP5 is not known to be regulated by STAT3 but our data suggested this, which we confirmed using an inhibitor of STAT3 (Stattic) that prevented IL11-induced *DUSP5* expression (Fig. 1g).

We then stimulated TECs with IL11 for 2 h in the presence or absence of Stattic and assessed signaling pathways and DUSP5 expression by western blotting. Inhibition of STAT3 activity prevented IL11-induced DUSP5 expression, which was associated with increased pERK expression, thus establishing the presence of an early onset STAT3-DUSP5-pERK feedback loop in TECs, which is not seen in other cells (Fig. 1h).

IL11 is pro-inflammatory in fibroblasts where it causes STAT3-dependent IL33 expression[25]. Further inspection of the RNA-seq data revealed that *IL33* is also highly induced (14.1-fold, $P = 1.7 \times 10^{-54}$) in TECs at 6 h post stimulation, along with *CCL20* (25.7-fold, $P = 9.9 \times 10^{-06}$) and *CXCL8* (16.5-fold, $P = 3.2 \times 10^{-55}$) (Supplementary Data 1), which are also increased in IL11-stimulated fibroblasts[25]. Some of the most downregulated genes following IL11 stimulation were TEC-specific channel genes (e.g., renal tubular urea transporter (*SLC14A2*) and aquaporin 6 (*AQP6*)), suggestive of TEC dedifferentiation during EMT (Supplementary Data 1).

### *Il11* null mice are protected from acute kidney injury

Having established that IL11 is upregulated in TECs during AKI, we subjected *Il11* null mice (*Il11^-/-^*)[29] or wild type controls to acute kidney injury and studied signaling effects, markers of pEMT as well as kidney morphology and function (Fig. 2a).

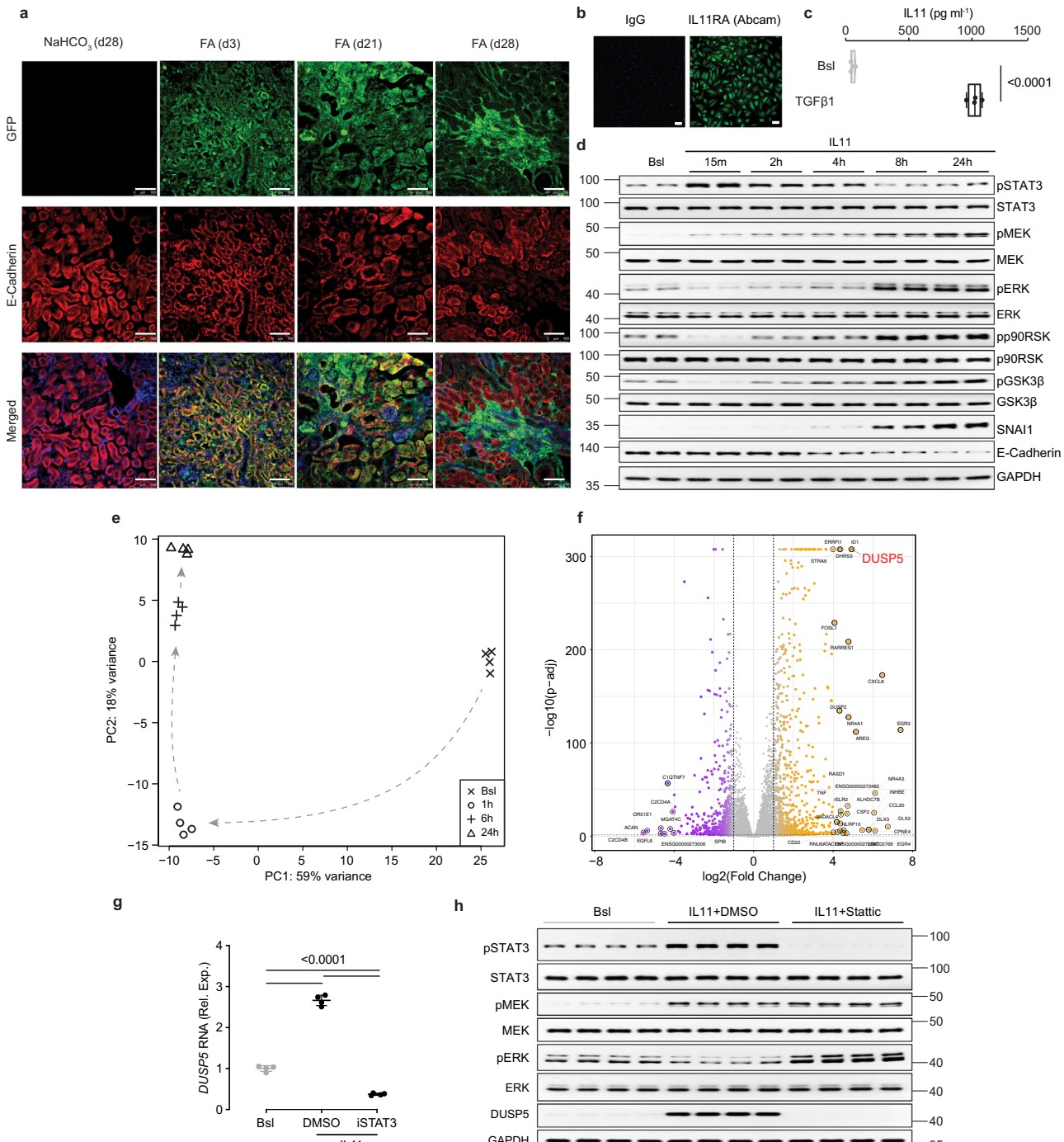

**Fig. 1 | Autocrine IL11 activity initiates a pro-inflammatory and mesenchymal program in tubular epithelial cells via activation of both STAT3 and ERK.**
**a** Representative immunofluorescence (IF) images (scale bars: 100 μm) of EGFP and E-Cadherin expression in the kidneys of $Il11^{EGFP/+}$ mice following folic acid (FA) injection (kidneys were collected at day 3, day 21, and day 28, representative dataset from n = 3/group. **b** Representative IF images of IL11RA staining of primary human renal TECs (scale bars: 100 μm; representative dataset from $n$ = 3/group). **c** ELISA of IL11 secretion from TGFβ1-stimulated TECs (24 h, $n$ = 4/group). **d** Western blots of STAT3, MEK, ERK, p90RSK, and GSK3β activation and expression levels of SNAI1 and E-Cadherin in IL11-stimulated TECs over a time course (representative dataset from $n$ = 4/group). **e, f** Data for RNA sequencing (RNA-seq) experiments on TEC stimulated with IL11 for 0 (Bsl), 1, 6, and 24 h. **e** Principal component analysis (PCA) of RNA-seq across the time-series. PC1 and PC2 account for 59% and 18% variance of gene expression, respectively. **f** Volcano plot displaying the adjusted $p$-value (−log10(p-adj)) and fold change (log2(Fold change)) of genes between IL11-stimulated cells versus baseline at 1-h time point. Differential expression analysis carried out using Wald test in DESeq2 and $p$-values adjusted using Benjamini & Hochberg method. Dashed lines are drawn to show log2(Fold change) value of −1 and +1 (vertical) and p-adj of 0.05 (horizontal). Upregulated, downregulated, and non-differentially expressed genes are labeled in orange, purple, and gray, respectively. DUSP5 is annotated in red for clarity. **g** Relative $DUSP5$ mRNA expression normalized to $GAPDH$ ($n$ = 4/group) and **h** Western blots showing STAT3, MEK, and ERK activation and DUSP5 expression ($n$ = 4/group) in IL11-stimulated TECs (2 h) in the presence of either DMSO (vehicle) or STAT3 inhibitor (Stattic). **a**–**h** IL11 (5 ng/ml), TGFβ1 (5 ng/ml), Stattic (2.5 μM). **c** Data are shown as box-and-whisker with median (middle line), 25th–75th percentiles (box), and minimum–maximum values (whiskers); two-tailed Student's $t$-test. **g** Data are shown as mean ± SD; one-way ANOVA with Tukey's correction. Bsl: Baseline; Rel. Exp: Relative expression. Source data are provided as a Source data file.

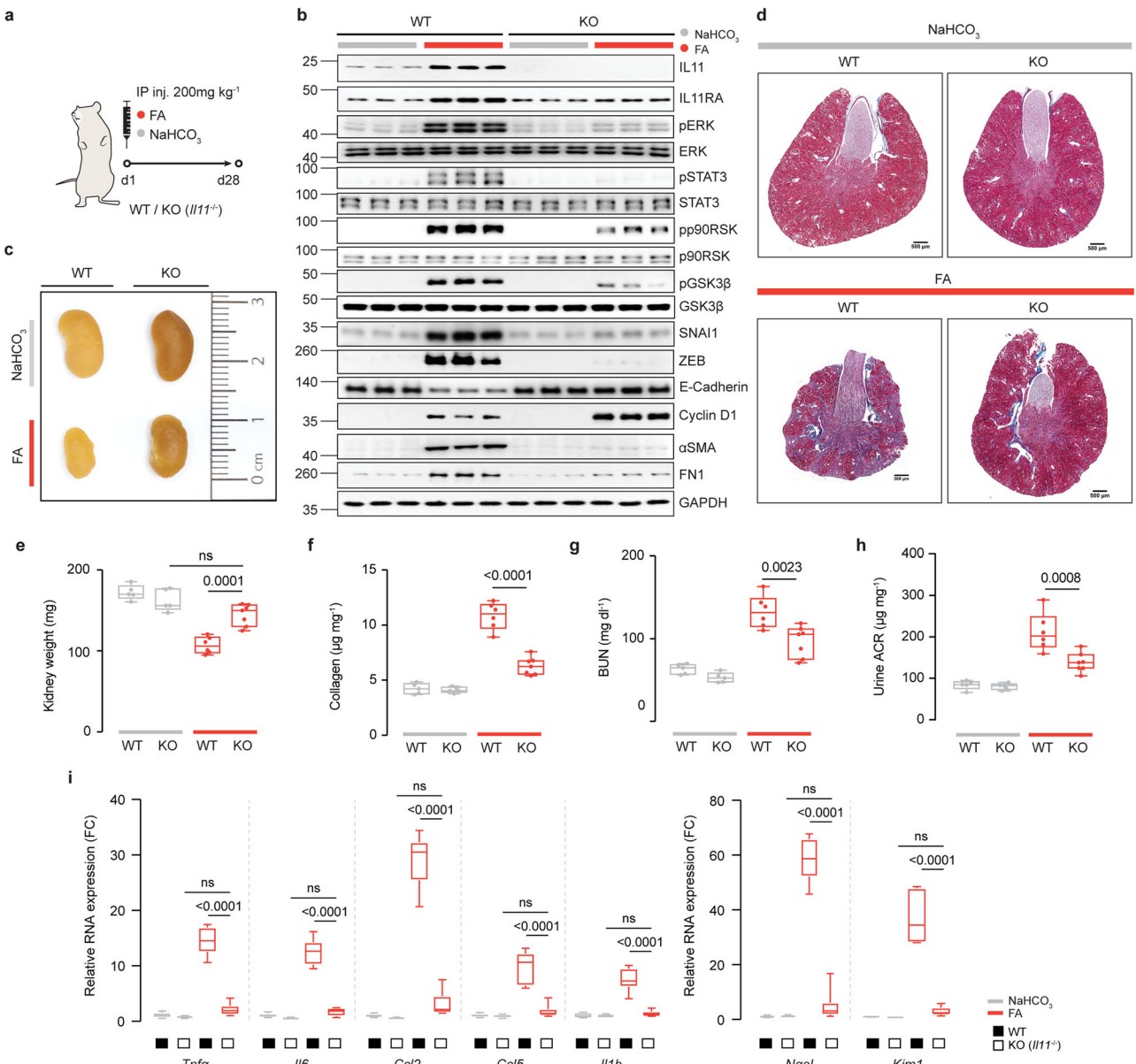

**Fig. 2 | *Il11* knockout mice are protected from epithelial–mesenchymal transition and renal pathology following acute kidney injury. a** Schematic of acute kidney injury (AKI) induction by FA administration in *Il11⁻/⁻* and WT mice for experiments shown in (**b**–**i**). A single dose of FA (200 mg/kg) or an equivalent volume of vehicle control (0.3 M NaHCO₃) was administered intraperitoneally, and the mice were sacrificed at D28. **b** Western blots showing renal expression of IL11, IL11RA1, pERK, ERK, pSTAT3, STAT3, pp90RSK, p90RSK, pGSK3β, GSK3β, SNAI1, ZEB, E-Cadherin, Cyclin D1, αSMA, Fibronectin (FN1), and GAPDH (representative dataset from *n* = 5/group), **c** representative kidney gross anatomy, **d** representative Masson's Trichrome images of whole albumin-to-creatinine ratios (ACRs), and **i** relative renal mRNA expression of kidney injury markers (*Ngal, Kim1*) and pro-inflammatory markers (*Tnfα, Il6, Ccl2, Ccl5, Il1β*). **e**–**i** Data are shown as box-and-whisker with median (middle line), 25th–75th percentiles (box), and minimum–maximum values (whiskers); WT-NaHCO₃, *Il11⁻/⁻* -NaHCO₃ (*n* = 5/group), WT-FA (*n* = 6), *Il11⁻/⁻* -FA (*n* = 7); two-way ANOVA with Sidak's correction. FC: Fold change. Source data are provided as a Source data file.

Following acute kidney injury, *Il11* mRNA and IL11 protein levels were increased in the kidneys of wild-type (WT) mice as compared to WT controls (Fig. 2b; Supplementary Fig. 2a, b). As expected, neither *Il11* mRNA nor IL11 were detected in *Il11⁻/⁻* mice, (Fig. 2b; Supplementary Fig. 2a, b). Twenty-eight days after FA-induced kidney injury, kidneys of WT mice were visibly smaller, weighed less, had higher collagen content, and were more fibrotic, as compared to *Il11⁻/⁻* mice (Fig. 2c–f; Supplementary Fig. 2c–e).

The mesenchymal markers αSMA and fibronectin (FN1), which are expressed in TECs undergoing pEMT and also in myofibroblasts, were upregulated in the injured kidneys from WT but not in *Il11⁻/⁻* mice (Fig. 2b). IL11RA1 expression was similar in WT and *Il11⁻/⁻* mice at

baseline, whereas it was increased in injured WT mice but not *Il11⁻/⁻* mice, showing evidence for IL11-induced IL11RA expression following kidney injury. As compared to WT controls, blood urea nitrogen (BUN) levels and urinary albumin-to-creatinine ratios (ACRs) were elevated in WT mice with AKI, which was significantly attenuated in *Il11⁻/⁻* mice (Fig. 2g, h).

In the injured kidneys of WT mice, there was increased pERK and pp90RSK and GSK3β inactivation (pGSK3β) along with upregulation in *Snai1*/SNAI1, *Zeb*/ZEB, Cyclin D1, and downregulation of E-Cadherin. All of these molecular phenotypes were mitigated in *Il11⁻/⁻* mice, except for Cyclin D1, which was increased further (Fig. 2b; Supplementary Fig. 2e).

STAT3-driven inflammation plays a key role in kidney injury[30] and IL11 causes TEC inflammation via STAT3 activation (Supplementary Fig. 1). We observed reduced levels of pSTAT3 and significantly attenuated inflammatory gene expression in injured kidneys of *ll11⁻/⁻* mice after FA, compared to controls (Fig. 2b, i). Markers of proximal renal tubule damage, *Ngal* and *Kim1* were strongly induced in WT kidneys after injury (58.4- and 36.8-fold, $P < 0.0001$; respectively). In contrast, *Ngal* and *Kim1* were not increased in injured kidneys of *ll11⁻/⁻* mice, as compared to uninjured WT or *ll11⁻/⁻* controls (Fig. 2i).

### Prevention of acute kidney injury using neutralizing IL11 antibodies

To determine whether a neutralizing IL11 antibody (X203)[24,31,32] impacts pEMT pathways and preserves kidney structure and function following severe AKI, we first explored effects of a range of X203 concentrations (0.5–10 mg/kg, as compared to IgG) on renal phenotypes in a preemptive administration mode (Fig. 3a). This X203 treatment regimen caused dose-dependent reductions in kidney collagen by both biochemical (hydroxyproline assay (HPA)) and histological analyses (Fig. 3b; Supplementary Fig. 3a, b) that was mirrored by improvement in kidney function reflected by dose-dependent reduction in BUN and serum creatinine levels (Fig. 2c; Supplementary Fig. 3c).

For replication, and to exclude potential off-target effects, we repeated the preemptive treatment experiment with a second, commercially available, neutralizing IL11 antibody (MAB218, R&D Systems) and compared its effects to X203 (Fig. 3d). X203 and MAB218 bind to different IL11 epitopes[26]. Confirming our dose range experiments, X203 reduced AKI-induced loss of kidney mass, limited renal fibrosis, and improved renal function while lowering inflammatory (*Tnfα*, *Il6*, *Ccl2*, *Ccl5*, *Il1β*), fibrosis (*Col1a1*, *Col1a2*, *Col3a1*, *Fn1*, *Acta2*) and tubular damage (*Kim1*, *Ngal*) markers (Fig. 3e; Supplementary Fig. 4a–f).

MAB218 is not as effective as X203 in inhibiting IL11 signaling[26] and was used at two concentrations, 10 mg/kg and 50 mg/kg, with IgG dosed accordingly as a control (Fig. 3d). Visually and quantitatively, the highest dose (50 mg/kg) of MAB218 preserved kidney mass while there was a non-significant trend toward preserving kidney mass with the lower dose (10 mg/kg) (Fig. 3e; Supplementary Fig. 4a). On the other hand, there were significant dose-dependent reductions in tubular damage, fibrosis, and inflammation with the highest dose of MAB218 also improving kidney function (Supplementary Fig. 4b–f). MAB218 also dose-dependently inhibited AKI-associated ERK activation, SNAI1 induction, E-Cadherin downregulation, and mesenchymal marker expression (Supplementary Fig. 4g).

### Anti-TGFβ causes inflammation following kidney injury whereas anti-IL11 does not

TGFβ is important for TEC pEMT, kidney fibrosis, and renal failure across kidney pathologies but TGFβ also has anti-inflammatory effects, which differs from IL11 that is pro-inflammatory (Supplementary Fig. 1)[25,33–35] Thus, we were interested in comparing the effects of X203 versus anti-TGFβ (clone 1D11) in our model of AKI.

Using the preemptive treatment regimen described above, we injected mice with either IgG, X203, or 1D11 on the day prior to kidney injury and then administered antibodies (20 mg/kg) twice per week for 28 days (Fig. 3f). At the end of the study, both X203 and 1D11 treatment reduced kidney fibrosis as assessed by HPA and histology analyses, but the magnitude of fibrosis reduction was greater with X203 (Fig. 3g–i).

Administration of 1D11 had no effect on BUN or ACR (Fig. 3j, k) in contrast to BUN and ACR levels in X203-treated mice that were similar to those seen in vehicle control mice (Fig. 3j, k). Both X203 and 1D11 reduced AKI-associated ERK activation and expression of fibrosis genes (*Col1a1*, *Col1a2*, *Col3a1*, *Fn1*, *Acta2*), but again, the magnitude of reduction was greater with X203. Notably, only X203 (and not 1D11) reduced the activation of pro-inflammatory STAT3 (Fig. 3l; Supplementary Fig. 5).

Mice with AKI receiving 1D11 had significant increases in the expression of markers of inflammation (*Tnfα*, *Il6*, *Ccl2*, and *Il1β*) and tubular damage (*Kim1* and *Ngal*), as compared to IgG-treated controls. In contrast, mice with AKI receiving X203 exhibited substantially reduced levels of inflammatory mediators and markers of tubular injury (Supplementary Figs. 4f and 5). Thus, while administration of either anti-IL11 or anti-TGFβ reduced fibrosis following AKI, anti-IL11 had singular effects to also attenuate inflammation and tubular damage while robustly preserving kidney function.

### Treatment of acute kidney injury with a neutralizing IL11 antibody

We next examined effects of X203 on long-term AKI outcomes using a treatment mode of administration given three days after kidney injury (Fig. 4a). Kidneys from mice receiving X203 appeared bigger than IgG controls, which was confirmed by kidney weights (Fig. 4b, c). In IgG control kidney extracts, immunoblotting showed FA injury-induced phosphorylation of ERK, p90RSK, and GSK3β that was associated with upregulation of mesenchymal markers (SNAI1, ZEB, αSMA, FN1), downregulation of E-Cadherin and a mild increase in Cyclin D1 (Fig. 4d). In contrast, all these pathways and protein levels were reduced in kidneys of FA-injured mice receiving X203, except for Cyclin D1, which was even more strongly induced (Fig. 4d).

As compared to those receiving IgG, X203-treated mice had less renal fibrosis and better renal function, as reflected in improved BUN, serum creatinine, and urinary ACR levels (Fig. 4d–j). Markers of tubule damage, pEMT, fibrosis, and inflammation were all diminished in X203-treated mice, as compared to IgG controls (Fig. 4k; Supplementary Fig. 6). Thus, our data suggest that anti-IL11 provides protection against chronic consequences of AKI even when administered three days after induction of kidney injury.

### Treatment of unilateral ureteral obstruction with a neutralizing L11 antibody

To examine the role of IL11 in a model of accelerated chronic kidney disease, we performed a separate set of experiments using the unilateral ureteral obstruction (UUO) model and first determined if IL11 was upregulated. Following UUO (10 days), obstructed kidneys of *ll11^{EGFP/+}* mice exhibited strong EGFP induction in TECs as compared to contralateral control kidneys (Fig. 5a). Western blotting confirmed high levels of IL11 expression after UUO (Fig. 5b).

We then examined the effects of anti-IL11 administered on days 4 and 8 post UUO, as compared to IgG control (Fig. 5c). In IgG-treated mice, UUO was associated with ERK, p90RSK, and GSK3β phosphorylation, upregulation of mesenchymal markers (SNAI1, ZEB, αSMA), downregulation of E-Cadherin, and a limited increase in Cyclin D1 (Fig. 5d). Similar to the FA model, these signaling and protein expression phenotypes were mitigated by X203, whereas again Cyclin D1 was yet further upregulated (Fig. 5d).

Levels of collagen content, extent of fibrosis, inflammation, and tubular damage were all significantly reduced in the obstructed kidneys from X203-treated mice, as compared to controls receiving IgG (Fig. 5e–h). Therefore, in UUO, a very different model of kidney injury to FA-induced AKI, IL11 is also upregulated in injured TECs and plays a central role in pEMT and fibrosis, which can be reversed by anti-IL11 administered after the inciting injury event.

### IL11 stimulates mesenchymal transition of human renal tubular epithelial cells

To explore the specific role of IL11 in renal TEC biology in greater detail and determine whether the molecular mechanisms observed in mice are relevant to human TECs, we performed a series of experiments in primary human renal TECs that are both a source and target of IL11 (Fig. 1b, c). The pro-fibrotic activity of TGFβ1 is dependent on autocrine IL11 signaling in stromal cells[15,31,36], and we examined if a similar

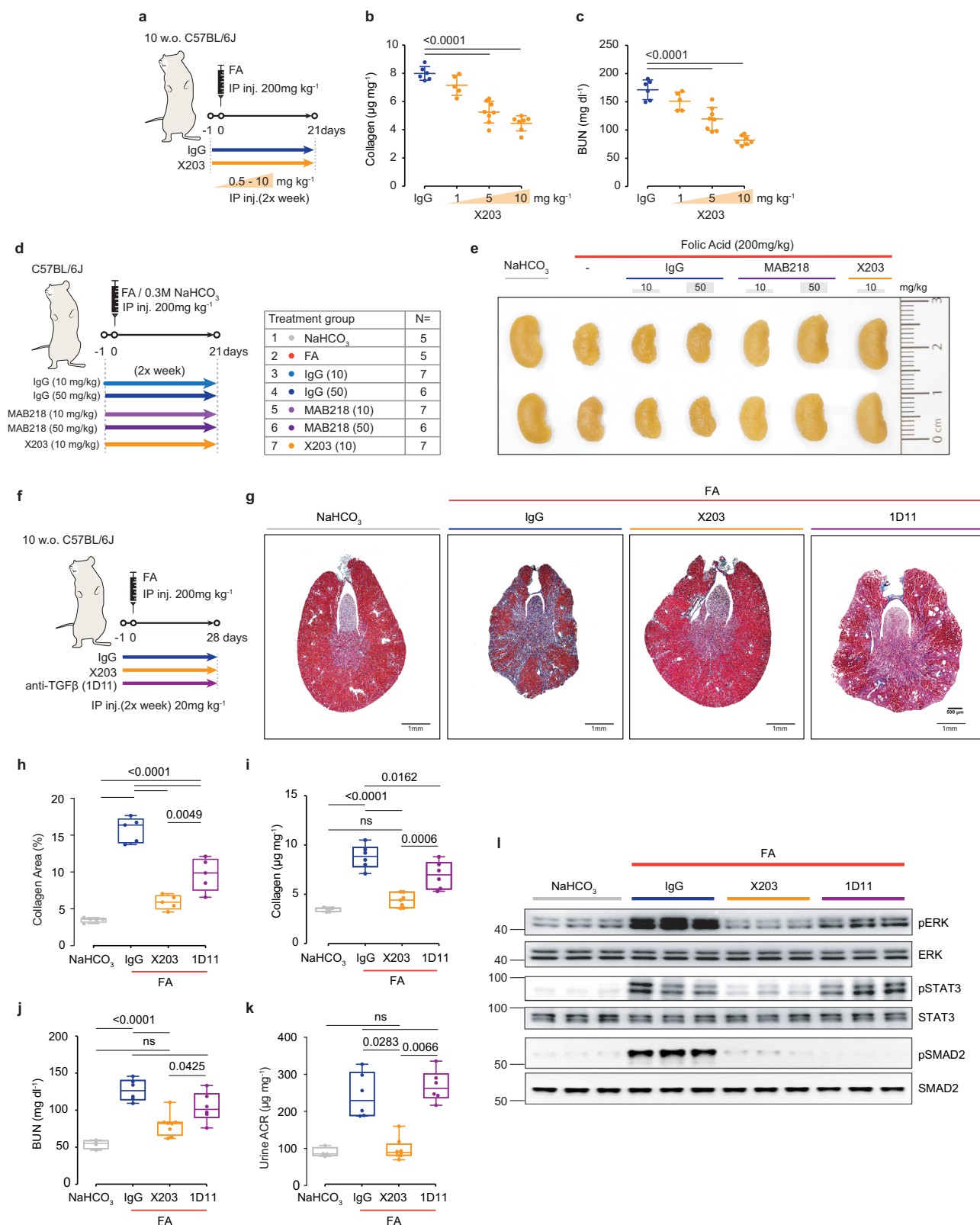

relationship might underlie TGFβ1-driven TEC pEMT. We observed that TGFβ1-induced αSMA, Collagen 1, and SNAI1 expression as well as collagen secretion were dependent, in part, on IL11 and that IL11 itself was sufficient to induce these phenotypes (Fig. 6a; Supplementary Fig. 7a, b).

At the signaling level, both TGFβ1 and IL11 increased pERK, pp90RSK, pGSK3β, SNAI1, ZEB, and αSMA expression and downregulated E-Cadherin, Cyclin D1, and PCNA (Fig. 6b). The effects of TGFβ1 on signaling and pEMT phenotypes were inhibited by X203 and also a neutralizing IL11RA antibody (X209[21,31]) but not IgG, indicating the induction of pEMT by TGFβ1 depends on IL11 in human TECs (Fig. 6b; Supplementary Fig. 7c). The primary importance of IL11-induced ERK activity in the p90RSK/GSK3β/SNAI1 cascade in IL11-stimulated TECs was shown using the MEK inhibitor, U0126 that

**Fig. 3 | Therapeutic comparison of X203, MAB218, and anti-TGFβ on renal phenotypes following acute kidney injury. a** Schematic of X203 dose-finding experiments for experiments shown in (**b**, **c**). IgG (11E10, 10 mg/kg) or X203 (1–10 mg/kg) was administered to mice 1 day before FA (200 mg/kg) injection. Mice were sacrificed at day 21 post FA (IgG (*n* = 6), X203 1 mg/kg (*n* = 5), X203 5 mg/kg (*n* = 8), X203 10 mg/kg (*n* = 7)). **b** Kidney collagen content by hydroxyproline assay and **c** BUN levels. **d** Schematic and **e** representative kidney gross anatomy for mice that were subjected to X203 and MAB218 treatment regimen following FA-mediated AKI. Mice were randomly divided into 7 groups with the number of mice/group indicated in the table. 10 or 50 mg/kg of IgG, 10 mg/kg of X203 or MAB218 (a commercial anti-IL11 from R&D Systems) were administered twice a week intraperitoneally starting from 1 day before FA injection until the mice were sacrificed (day 21). **f** Schematic of therapeutic comparison of X203 and anti-TGFβ: 20 mg/kg IgG (11E10), X203, or anti-TGFβ (1D11) was administered to mice 1 day before FA (200 mg/kg) injection for experiments shown in (**g–l**). Mice were sacrificed on day 28 post FA. **g** Representative Masson's Trichrome images of whole kidney cross section (scale bars: 500 μm, representative dataset from *n* = 5/group), **h** quantification of collagen area from Masson's Trichrome-stained kidney sections (*n* = 5/group), **i** kidney collagen content by hydroxyproline assay (NaHCO₃, FA + IgG, FA + 1D11 (*n* = 6/group), FA + X203 (*n* = 7)), **j** BUN, **k** urine ACR. **j**, **k** NaHCO₃ (*n* = 4), FA + IgG, FA + 1D11 (*n* = 6/group), FA + X203 (*n* = 8). **l** Western blots of ERK, STAT3, and SMAD2 activation (representative dataset from *n* = 5/group). **b**, **c** Data are shown as mean ± SD, **h–k** data are shown as box-and-whisker with median (middle line), 25th–75th percentiles (box), and minimum–maximum values (whiskers). **b**, **c** One-way ANOVA with Dunnett's correction, **h**, **i** one-way ANOVA with Tukey's correction, **k** Kruskal–Wallis with Dunn's correction. Source data are provided as a Source data file.

inhibited all downstream signaling events as well as IL11-dependent pEMT phenotypes (Supplementary Fig. 7d–g).

In fibroblasts, TGFβ1 most strongly induces *IL11*[15,31] but IL11 secretion also occurs following exposure of cells to a variety of pathogenic stimuli that include viruses, cytokines, chemokines, peptides, and reactive oxygen species[15]. We therefore stimulated TECs with factors thought important for pEMT and kidney disease: angiotensin II (AngII), basic fibroblast growth factor (bFGF), endothelin 1 (ET-1), reactive oxygen species ($H_2O_2$), platelet-derived growth factor (PDGF), or TGFβ1.

We found that all of these factors induced IL11 secretion from TECs, albeit to variable degrees, (Supplementary Fig. 8a) and also upregulated Collagen 1, SNAI1, and αSMA (Fig. 6c, d; Supplementary Fig. 8b–d). X203 blocked induction of Collagen 1, SNAI1, and αSMA by all factors (Fig. 6c, d; Supplementary Fig. 8b–d) and prevented other downstream effects including activation of ERK/p90RSK, inhibition of GSK3β, increased mesenchymal marker expression, and reduced E-Cadherin and cell proliferation markers (Cyclin D1 and PCNA) (Fig. 6d), suggesting a key role for IL11 in pEMT induction by a range of pathological factors.

It is established that SNAI1 upregulation is critical for TGFβ1-induced pEMT. We showed that TGFβ1-induced upregulation of SNAI1 in TECs is IL11-dependent and demonstrated the mechanistic importance of SNAI1 expression for both IL11 and TGFβ1 effects on TEC pEMT using siRNA (Supplementary Fig. 8e). Our data suggest a general dependency of the IL11-driven ERK/p90RSK/GSK3β/SNAI1 axis for pEMT of human TECs and we summarize our findings in a cartoon (Fig. 6f).

## Paracrine IL11 activity from TECs activates renal fibroblasts

Our time course studies of IL11 expression in *Il11^EGFP/+* mice with AKI suggested that IL11 is first upregulated in TECs and then the stroma, which becomes the dominant source later in disease (Fig. 1a). This implies that IL11 secreted from TECs has autocrine activity in TECs and may then have paracrine actions on resident stromal cells, as seen for hepatocytes and hepatic stellate cells in the injured liver[37]. To examine this possibility, we incubated primary human renal fibroblasts with conditioned TEC media from control or TGFβ1-stimulated TECs in the presence of anti-TGFβ (1D11) to exclude a TGFβ effect, 1D11 + IgG, or 1D11 + X203 for 24 h (Fig. 6g).

We found that media from TGFβ1-stimulated TECs induced fibroblasts-to-myofibroblast transformation and that neither IgG nor 1D11 had any effect on this transition whereas X203 prevented it (Fig. 6h; Supplementary Fig. 9). These data show that paracrine activity from injured TECs on myofibroblasts is IL11-dependent and not directly related to TGFβ1 activity, in keeping with data from an early study[38].

## Deletion of *Il11ra1* in renal TECs prevents long-term sequelae of AKI

Our data suggested that a major initial activity for IL11 in AKI is in TEC mesenchymal transition and it follows that inhibition of IL11 signaling

in TECs should protect against kidney injury. To test this possibility, we deleted *Il11ra1* from TECs by crossing mice bearing *LoxP*-targeted *Il11ra1* alleles[21] with Cadherin 16 (*Cdh16*)-driven Cre transgenic mice[39], which express Cre specifically in tubular epithelia, to generate mice with conditional knockout (CKO) of *Il11ra1* in TECs. These mice were then used in the FA AKI model (Fig. 7a).

At baseline, kidneys from CKO mice appeared normal, as expected, given mice globally null for *Il11ra1*[15] or *Il11* have normal kidney structure and function (Fig. 2c, d, g, h)

Immunoblotting of whole kidney extracts CKO mice showed depletion of IL11RA1 consistent with TECs being a sizable reservoir of IL11RA1 (Fig. 7b), which was confirmed by immunohistochemistry analysis (Supplementary Fig. 10a). IL11RA1 was increased in injured WT kidneys (as seen in WT controls for *Il11^-/-* mice (Fig. 2b)) but not in injured CKO kidneys (Fig. 7b). Thus IL11-induced IL11RA1 expression in the damaged kidney is dependent on intact IL11 signaling in TECs.

Twenty-eight days following FA administration, WT control mice had the expected changes of increased pERK, pp90RSK, and pGSK3β levels that was associated with greater expression of SNAI1, FN1, ZEB, and αSMA along with reduced E-Cadherin. These signaling events and the reciprocal changes in mesenchymal and epithelial markers were diminished in kidneys of CKO mice following AKI. Cyclin D1 levels, which were mildly increased in WT kidneys after FA, were further increased in FA-injured CKO kidneys (Fig. 7b).

At the study end-point, WT mice had lower kidney weights than those receiving buffer, whereas kidney mass of injured CKO mice was preserved (Fig. 7c). Injured WT mice had fibrotic kidneys and impaired kidney function whereas the CKO mice had similar levels of renal collagen content, BUN, serum creatinine, and urinary ACR to those of uninjured WT or CKO mice (Fig. 7d–i).

At the gene expression level, there was upregulation of fibrosis (range: 6–26-fold), inflammation (10–30-fold), and tubule injury genes (50–100-fold) in WT mice following injury (Fig. 7j; Supplementary Fig. 10b). In contrast, the expression levels of these markers were similar between uninjured WT, uninjured CKO mice and injured CKO kidneys (Fig. 7j; Supplementary Fig. 10b). Taken together, these data show that autocrine IL11-driven pEMT in damaged TECs is a major initiating determinant of renal impairment and fibrosis following AKI.

## Anti-IL11 reverses renal epithelial cell mesenchymal transition

We next addressed whether anti-IL11 could reverse TEC pEMT, rather than just prevent it (Fig. 6a–d). We stimulated primary human renal TECs with TGFβ1 for 72 h, followed by 24 h of further stimulation but with the addition of either IgG or X203 (Fig. 8a).

As compared to IgG control, X203 reversed phosphorylation of ERK, p90RSK, and GSK3β, diminished the expression of pEMT markers (Collagen, SNAI1, ZEB, FN1, αSMA) and upregulated E-Cadherin, despite ongoing TGFβ1 stimulation (Fig. 8a, b; Supplementary Fig. 11).

Withdrawal of TECs from the cell cycle is a particular feature of SNAI1-driven pEMT that prevents regeneration[1,5,40] and it was notable

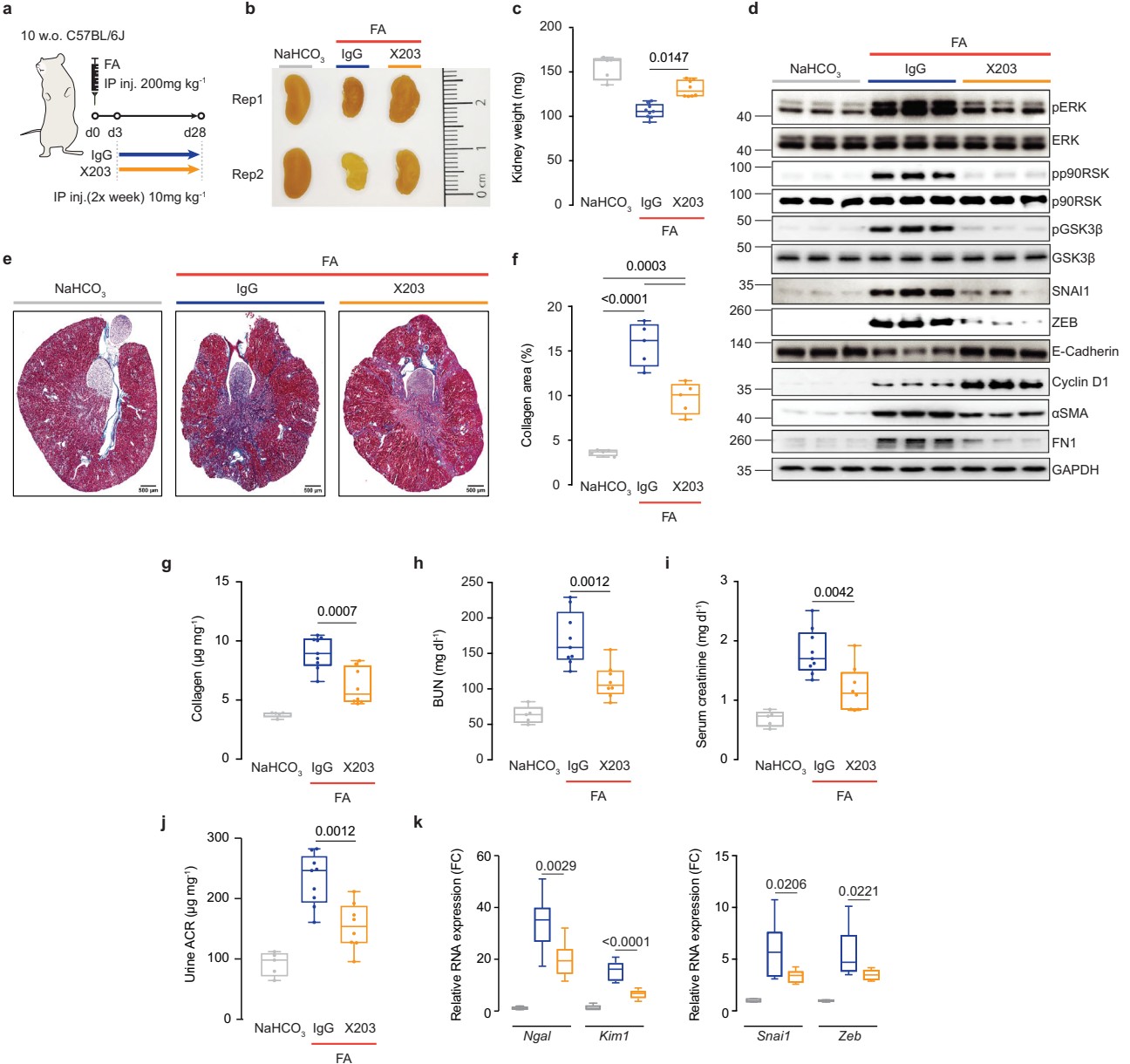

**Fig. 4 | Anti-IL11 reduces tubular damage, renal fibrosis and preserves kidney function following folic acid-induced acute kidney injury. a** Schematic of X203 therapeutic dosing of FA-injured mice for experiments shown in (**b**–**k**). IgG/X203 (10 mg/kg) was administered biweekly starting from day 3 following FA injury. Blood and kidney phenotypes were analyzed on day 28. **b** Representative kidney gross anatomy, **c** kidney weights, **d** Western blots of pERK, ERK, pp90RSK, p90RSK, pGSK3β, GSK3β, SNAI1, ZEB, E-Cadherin, Cyclin D1, αSMA, FN1, and GAPDH (representative dataset from $n = 5$/group), **e** representative Masson's Trichrome images of whole kidney cross section (scale bars: 500 μm, representative dataset from $n = 5$/group), **f** quantification of collagen area from Masson's Trichrome-stained kidney sections ($n = 5$/group), **g** kidney collagen contents by hydroxyproline assay, the levels of **h** BUN, **i** serum creatinine, and **j** urine ACRs, **k** relative mRNA expression of kidney injury (*Ngal*, *Kim1*) and pEMT (*Snai1*, *Zeb*) markers. **c**, **f**–**k** Data are shown as box-and-whisker with median (middle line), 25th–75th percentiles (box), and minimum–maximum values (whiskers). **c**, **g**–**k** NaHCO3 ($n = 5$), FA + IgG ($n = 9$), FA + X203 ($n = 8$). **c** Kruskal–Wallis with Dunn's correction, **f**–**k** one-way ANOVA with Tukey's correction. FC: Fold change. Source data are provided as a Source data file.

that the number of EDU[+ve] TECs and the expression levels of Cyclin D1 and PCNA were restored to baseline levels in X203-treated TECs (Fig. 8a, c, d).

## A neutralizing IL11 antibody reverses chronic kidney disease

We demonstrated that treatment with anti-IL11 early in the course of two distinct models of AKI prevents subsequent fibrosis and chronic injury responses (Figs. 3–5). As a next step, we tested the efficacy of anti-IL11 treatment in kidneys with established structural and functional changes characteristic of human CKD. In the FA model, by three weeks after injury kidneys are severely damaged with extensive fibrosis

and reduced function. Therefore, we studied the impact of 12 weeks of treatment with X203 starting at three weeks after FA-induced AKI (Fig. 9a).

At baseline (three weeks post injury), kidney weights were reduced to ~67% of their initial size and this was unchanged by 12 weeks of IgG administration (Fig. 9b, c). In contrast, there was a significant and progressive increase in kidney mass over the course of the study period with X203 therapy (Fig. 9c). At the end of the experiment, X203-treated mice had regained ~50% of the lost kidney mass (Fig. 9c). Over the same period, kidney collagen content and the extent of histological fibrosis was progressively diminished with X203 therapy, but these

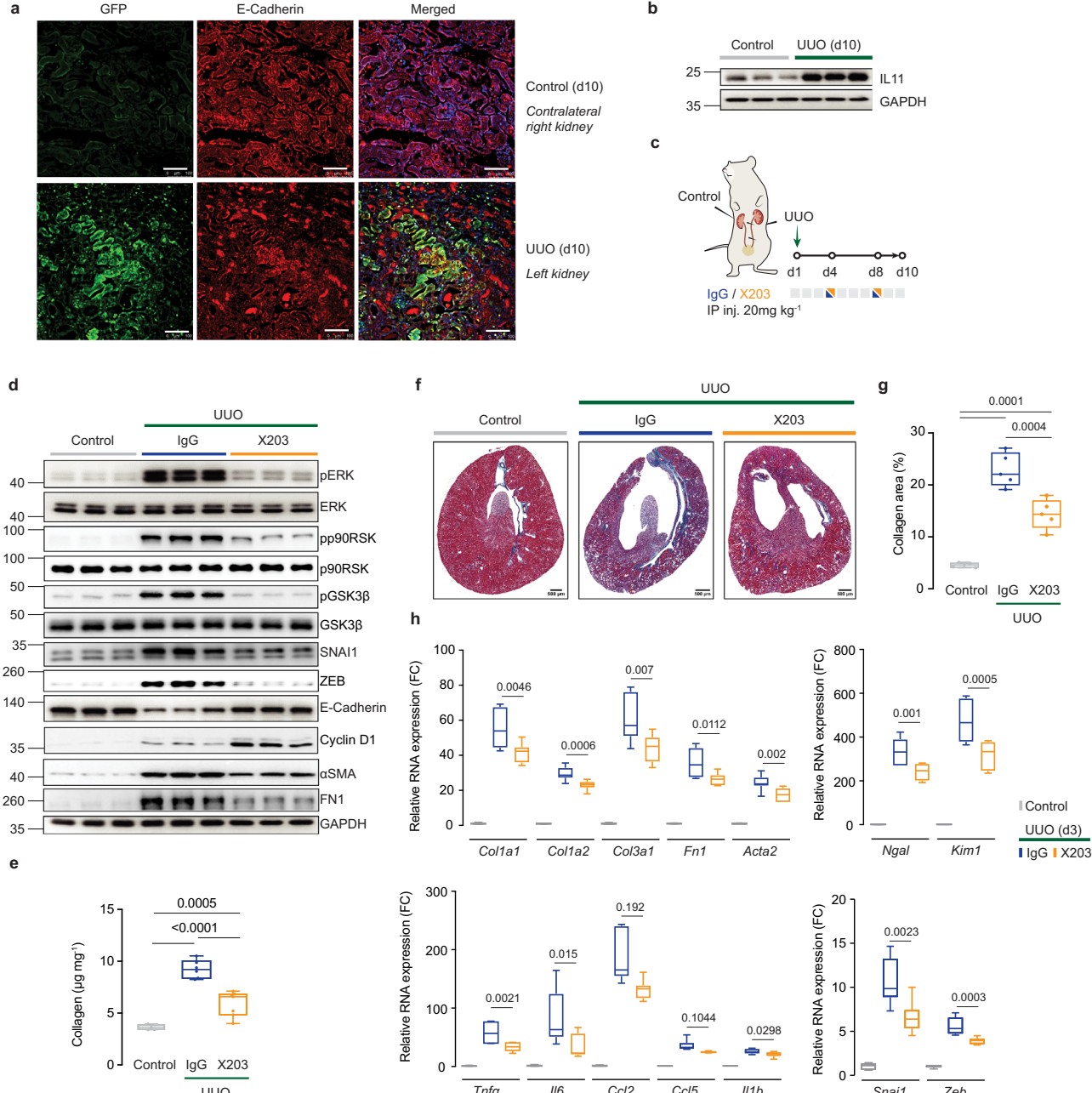

**Fig. 5 | Therapeutic targeting of IL11 protects mice from UUO-induced renal tubule damage, inflammation, and fibrosis. a** Representative immuno-fluorescence images (scale bars: 100 μm) of EGFP and E-Cadherin expression in UUO and control kidneys of *IL11*^EGFP/+ mice (representative dataset from *n* = 3/group) and **b** Western blots of IL11 protein expression in control and UUO kidneys (representative dataset from *n* = 5/group). Kidneys were collected at day 10 and contralateral kidneys were used as controls. **c** Schematic of X203 therapeutic dosing of mice subjected to left unilateral ureteral obstruction for experiments shown in (**d–h**). IgG/X203 (10 mg/kg) was administered on day 4 and day 8 post UUO and kidneys were collected on day 10; contralateral (right) kidneys from the IgG group were used as controls. **d** Western blots of pERK, ERK, pp90RSK, p90RSK, pGSK3β, GSK3β, SNAI1, ZEB, E-Cadherin, Cyclin D1, αSMA, FN1, and GAPDH

(representative dataset from *n* = 5/group), **e** kidney collagen content by hydro-xyproline assay (control, UUO + X203 (*n* = 7/group), UUO + IgG (*n* = 6)), **f** representative Masson's Trichrome images of whole kidney cross section (scale bars: 500 μm, representative dataset from *n* = 5/group), **g** quantification of collagen area from Masson's Trichrome-stained kidney sections (*n* = 5/group), and **h** relative renal mRNA expression of pro-inflammatory markers (*Tnfα, Il6, Ccl2, Ccl5, Il1β*), fibrosis markers (*Col1a1, Col1a2, Col3a1, Fn1, Acta2*), and kidney injury markers (*Ngal, Kim1*) (*n* = 7/group). **e, g, h** Data are shown as box-and-whisker with median (middle line), 25th–75th percentiles (box), and minimum–maximum values (whis-kers); one-way ANOVA with Tukey's correction except for (**h**, *Ccl2* and *Ccl5*) which were analyzed by Kruskal–Wallis with Dunn's correction. FC: Fold change. Source data are provided as a Source data file.

parameters were unaffected by IgG administration (Fig. 9d, e; Sup-plementary Fig. 12a).

In mice with chronic kidney injury, X203 reversed phosphoryla-tion of ERK, p90RSK, and GSK3β, reduced the expression of pEMT markers (SNAI1, ZEB, FN, αSMA), restored E-Cadherin expression, and increased proliferation markers (Fig. 9f). In addition, X203 significantly improved BUN levels, serum creatinine, and urinary ACR levels by the study end (Fig. 9g–i). Fibrosis, inflammation, tubule injury, and pEMT mRNA expression levels were also reduced with X203 treatment, as compared to IgG controls (Fig. 9j and Supplementary Fig. 12b).

To assess the cellular compartment(s) associated with the regenerative features seen in X203-treated mice we administered

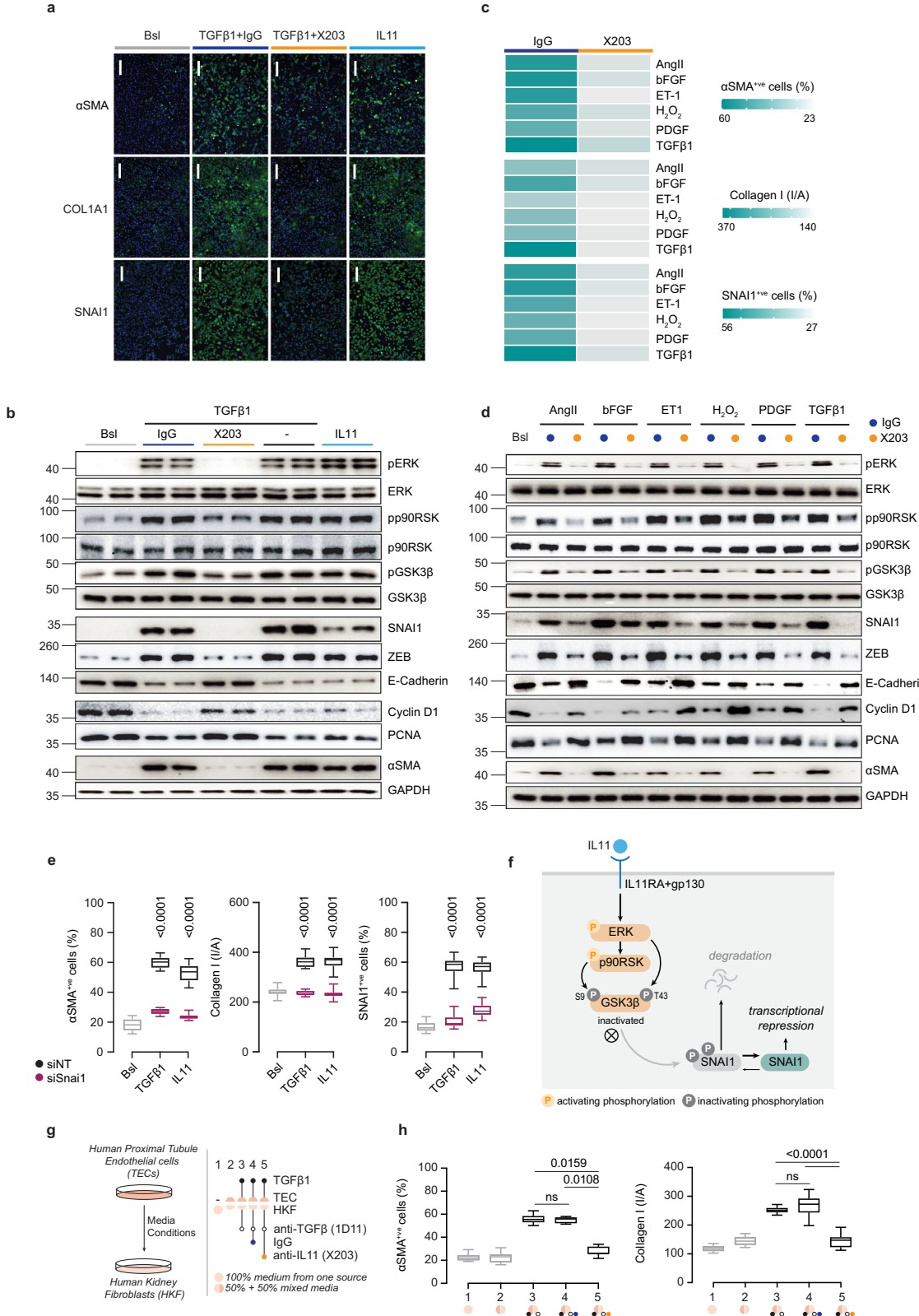

EdU to mice 6 weeks after antibody therapy was started (9 weeks after AKI). This revealed significantly more EdU[+ve] incorporation to TECs in kidneys from X203-treated mice as compared to IgG or vehicle control (Fig. 6k, l). Thus, in the setting of established chronic kidney disease, anti-IL11 is permissive for TEC regeneration, reverses fibrosis and pEMT phenotypes, and improves renal function.

## Discussion

In response to severe kidney injury, TECs release a variety of factors that cause autocrine SNAI1-dependent TEC dysfunction and proliferative failure, while activating myofibroblasts and inflammatory cells in paracrine[1,5,6,41]. Our data show that mesenchymal transition of TECs is critically dependent on IL11. Notably, TECs express IL11RA and secrete IL11 in response to injury in vivo or stimulation by a range of

**Fig. 6 | Autocrine IL11 activity causes mesenchymal transition of human renal TECs and paracrine IL11 activity from TECs stimulates myofibroblasts.**
**a** Representative IF images of αSMA[+ve] and SNAI1[+ve] cells and Collagen 1 immunostaining (scale bars: 100 μm, representative dataset from n = 3/group) and
**b** Western blots of pERK, ERK, pp90RSK, p90RSK, pGSK3β, GSK3β, SNAI1, ZEB, E-Cadherin, PCNA, Cyclin D1, αSMA, and GAPDH (representative dataset from n = 4/group) from TECs stimulated with either IL11 or TGFβ in the presence of IgG/X203.
**c** Heatmaps showing quantification of αSMA[+ve] and SNAI1[+ve] cells and Collagen 1 intensity/area (n = 14/group; values are shown in Supplementary Fig. 8c) and
**d** Western blots of pERK, ERK, pp90RSK, p90RSK, pGSK3β, GSK3β, SNAI1, ZEB, E-Cadherin, PCNA, Cyclin D1, αSMA, and GAPDH for TECs stimulated with various pro-fibrotic factors in the presence of IgG/X203 (representative dataset from n = 4/group). **e** Quantification of αSMA[+ve] and SNAI1[+ve] cells and Collagen 1 intensity/area from TECs stimulated with TGFβ or IL11 subjected to siRNA knockdown for SNAI1

(NT: non-targeting siRNA control) (24 h; n = 14/group; Bsl, baseline). **f** Schematic showing mechanism and signaling pathway by which IL11 induces repression of SNAI1 transcription. **g** Schematic and **h** quantification of αSMA[+ve] cells and Collagen 1 immunostaining of HKFs (n = 14/group) for media transfer experiments in which conditioned media from control or TGFβ-stimulated TECs (24 h) were used to treat primary human kidney fibroblasts (HKFs) in the presence of anti-TGFβ (1D11) alone or with either IgG or X203. **a**–**h** AngII (100 nM), bFGF (10 ng/ml), Endothelin 1 (ET-1, 250 ng/ml), H2O2 (0.2 mM), IL11 (5 ng/ml), PDGF (20 ng/ml), TGFβ1 (5 ng/ml), IgG (2 μg/ml), X203 (2 μg/ml), anti-TGFβ (clone 1D11, 2 μg/ml), siNT/siSNAI1 (25 nM); 24-h stimulation. **e, h** Data are shown as box-and-whisker with median (middle line), 25th–75th percentiles (box), and minimum–maximum values (whiskers); one-way ANOVA with Tukey's correction (except for H, αSMA[+ve] which was analyzed by Kruskal–Wallis with Dunn's correction). Bsl: Baseline; I/A: Intensity per area. Source data are provided as a Source data file.

pathological mediators in vitro, indicating that this common cellular program can be triggered in response to injury from diverse causes.

In TECs, IL11 activates ERK and p90RSK to phosphorylate GSK3β (Thr43 and Ser9, respectively) resulting in GSK3β inactivation[10]. This prevents GSK3β-mediated SNAI1 phosphorylation causing SNAI1 accumulation and the transcriptional suppression of its target genes, prototypically E-Cadherin[7,9]. Interestingly, SNAI1 is increased in diseased glomerular podocytes, suggesting a role for pEMT, and perhaps IL11, in podocyte dysfunction[16,42,43]. GSK3β-dependent SNAI1 degradation is also required for liver regeneration[44,45] and this pathway, and IL11, may be more generally relevant[21,46].

We found that neutralizing IL11 antibodies can prevent the long-term consequences of severe AKI including loss of kidney mass, increased fibrosis, and reduced kidney function. This intervention was effective in two distinct experimental models and beneficial dose-dependent effects were observed whether treatment was given before or after acute injury. Strikingly, we found that administration of anti-IL11 caused reversal and resolution of established chronic structural and functional changes in a mouse model of CKD. This was associated with TEC proliferation and restoration of kidney parenchymal mass, providing evidence that therapy-enabled kidney regeneration is possible.

Inflammation is a central component of kidney dysfunction in AKI and CKD[47,48]. While TGFβ1 is an important cause of TEC pEMT and renal fibrosis, it has notable anti-inflammatory activities[33,38,49]. Consequently, and unfortunately, dose-limiting on-target toxicities linked to inflammation have thwarted anti-TGFβ clinical trials in CKD[13]. We compared anti-TGFβ with anti-IL11 in AKI and found both interventions to reduce fibrosis but only anti-IL11 improved renal function. While anti-IL11 reduced renal inflammation and tubule damage, anti-TGFβ did the opposite: exacerbating kidney inflammation and tubular injury. The anti-inflammatory effect associated with IL11 inhibition is in keeping with its direct, stat-dependent effect on pro-inflammatory gene expression (e.g., *IL33, CCL20, CXCL8*) in TECs, which is similarly seen in fibroblasts[25].

In conclusion, anti-IL11 prevents, treats, and reverses kidney dysfunction across mouse models of AKI and CKD. The molecular mechanisms underlying these effects relate to reduced IL11-mediated ERK/p90RSK activation and SNAI1 accumulation in TECs. Intriguing, IL11-stimulated ERK/p90RSK activation was similarly shown to dually phosphorylate and inactivate LKB1, which may additionally be relevant for IL11 effects in kidney disease[18,50]. In CKD, anti-IL11 was associated with TEC proliferation, restoration of kidney parenchymal mass and reversal of renal failure. Importantly, unlike anti-TGFβ, anti-IL11 therapy reduces renal inflammation/tubule damage and target safety signals in humans and mice deleted for *IL11RA* or *IL11* are encouraging[35]. Our studies suggest potential therapeutic opportunities for anti-IL11 in acute and chronic kidney diseases.

## Methods

### Antibodies
Commercial antibodies: Collagen I (ab34710, Abcam), Cyclin D1 (55506, clone E3P5S, CST), E-Cadherin (3195, clone 24E10, CST), DUSP5 (ab200708, clone EPR19684, Abcam), pERK1/2 (4370, clone D13.14.4E, CST), ERK1/2 (4695, clone 137F5, CST), Fibronectin (ab2413, Abcam), GAPDH (2118, clone 14C10, CST), GFP (ab6673, Abcam), pGSK3β (5558, clone D85E12, CST), GSK3β (12456, clone D5C5Z, CST), commercial anti-IL11 (MAB218, clone 22626, R&D Systems), IL11RA (ab125015, clone EPR5446, Abcam; IF), pMEK1/2 (9154, clone 41G9, CST), MEK1/2 (4694, clone L38C12, CST), pp90RSK (11989, clone D3H11, CST), RSK (9355, clone 32D7, CST), PCNA (13110, clone D3H8P, CST), αSMA (ab7817, clone 1A4, Abcam; IF), αSMA (19245, clone D4K9N, CST; WB), SNAI1 (ab180714, Abcam; IF), SNAI1 (3879, clone C15D3, CST, WB), pSTAT3 (4113, clone M9C6, CST), STAT3 (4904, clone 79D7, CST), neutralizing anti-TGFβ (clone 1D11, Aldevron), ZEB (70512, clone E2G6Y, CST), anti-rabbit HRP (7074, CST), anti-mouse HRP (7076, CST), anti-rabbit Alexa Fluor 488 (ab150077, Abcam), anti-rabbit Alexa Fluor 647 (ab150079, Abcam), anti-mouse Alexa Fluor 488 (ab150113, Abcam), anti-goat Alexa Fluor 488 (ab150129, Abcam). All commercially available antibodies have been validated by their manufacturer as indicated in their respective datasheet and/or website.

Custom antibodies: IgG (clone 11E10), neutralizing anti-IL11 (clone X203), neutralizing anti-IL11RA (clone X209; neutralizing study and IHC), and neutralizing anti-TGFβ were manufactured by Aldevron and have been validated previously. IgG (11E10) was validated for its non-effect on human/mouse cells and hence its suitability as a control antibody[21]. X203 was validated for neutralization of human and mouse IL11[21,31], for Western Blot[31], and for IHC (in this manuscript). X209 was validated for neutralization of human and mouse IL11RA[31], for IHC[16], and for WB (in this manuscript). 1D11 was validated for neutralization of human and mouse TGFβ[51,52].

Dilution for each antibody is provided in the respective method section in which the antibody is used and in the reporting summary.

### Recombinant proteins
Commercial recombinant proteins: Human angiotensin II (A9525, Sigma-Aldrich), human bFGF (233-FB-025, R&D Systems), human endothelin (ET-1, 1160/100U, TOcris Bioscience), human PDGF (220-BB-010, R&D Systems), and human TGFβ1 (PHP143B, Bio-Rad).

Custom recombinant proteins: Recombinant human IL11 (rhIL11, UniProtKB:P20809) was synthesized by GenScript using a mammalian expression system.

### Chemicals
Bovine serum albumin (BSA, A7906, Sigma), 16% Formaldehyde (w/v), methanol-free (Cat #28908, Pierce™ ThermoFisher Scientific), DAPI (D1306, ThermoFisher Scientific), Hydrogen peroxide solution (H2O2, 323381, Sigma), Stattic (S7947, Sigma), U0126 (9930, CST), Triton X-100 (T8787, Sigma), Tween-20 (170-6531, Bio-Rad).

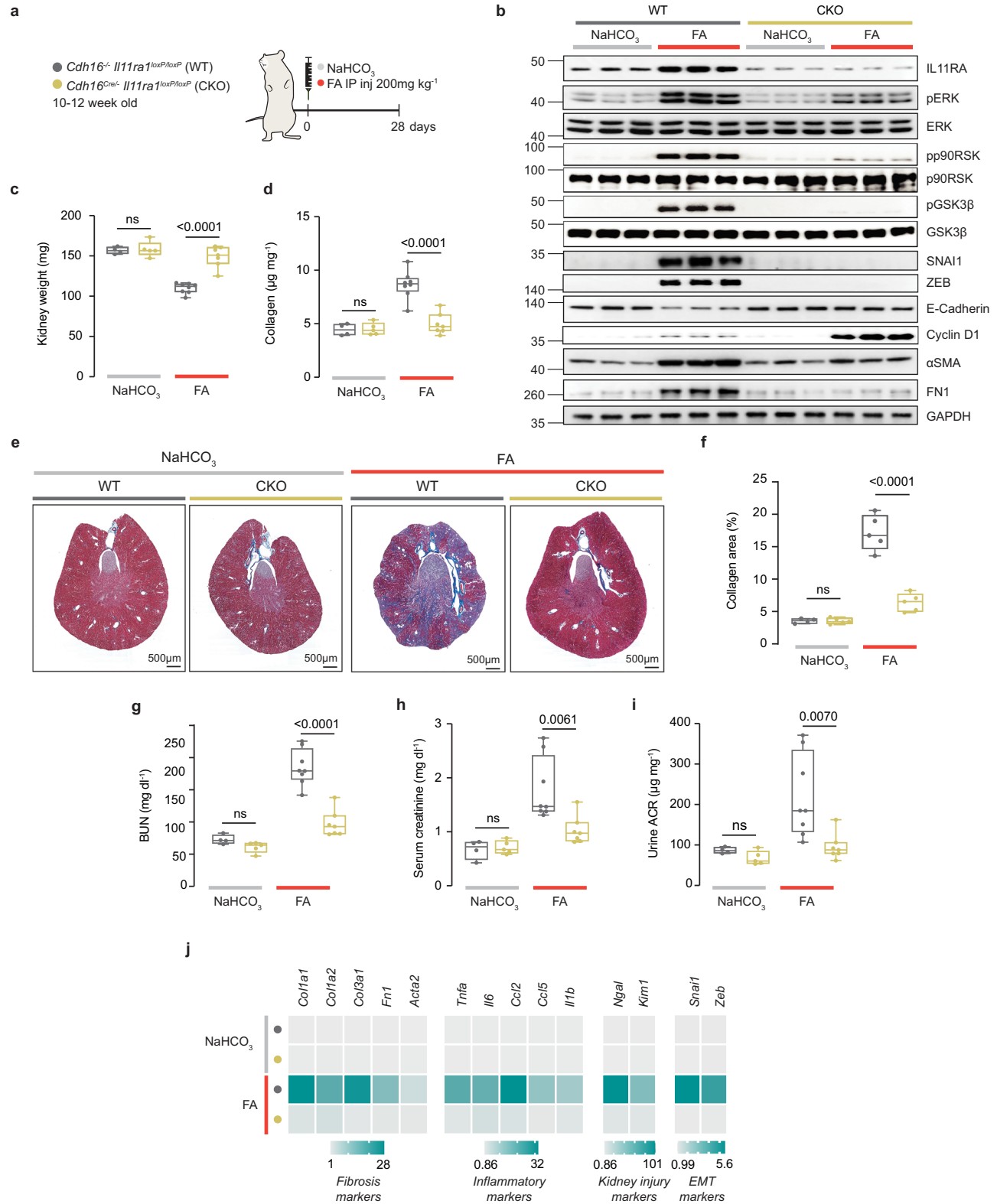

## Ethics statements

All experimental protocols involving human subjects (commercial primary human cell lines) have been performed in accordance with the *ICH Guidelines for Good Clinical Practice*. As written in their respective datasheets, ethical approvals have been obtained by the relevant parties and all participants gave written informed consent to either ScienCell or InnoProt from which primary human renal tubular epithelial cells or primary human kidney fibroblasts, respectively were commercially sourced.

Animal studies were carried out in compliance with the recommendations in the Guidelines on the Care and Use of Animals for Scientific Purposes of the National Advisory Committee for Laboratory Animal Research (NACLAR). All experimental procedures were approved (SHS/2019/1482 and SHS/2019/1483) and conducted in

**Fig. 7 | Deletion of *Il11ra1* in tubular epithelial cells protects against acute kidney injury. a** Schematic of induction of AKI injury in Cadh16-Cre[+/-] *Il11ra1[loxP/loxP]* mice for experiments shown in (**b–i**). Cadh16-Cre[+/-] *Il11ra1[loxP/loxP]* and littermate control (Cadh16-Cre[-/-] *Il11ra1[loxP/loxP]*) mice were injected intraperitoneally with 200 mg/kg FA. Blood and kidneys were collected on day 28 post FA. **b** Western blots showing renal expression of IL11, pERK, ERK, pp90RSK, p90RSK, pGSK3β, GSK3β, SNAI1, ZEB, E-Cadherin, Cyclin D1, αSMA, Fibronectin (FN1), and GAPDH (representative dataset from *n* = 5/group, except for WT-NaHCO₃: *n* = 4), **c** kidney weights, **d** kidney collagen content by hydroxyproline assay, **e** representative Masson's Trichrome images of whole kidney cross section (scale bars: 500 μm, representative dataset from *n* = 5/group, except for WT-NaHCO₃: *n* = 4),

**f** quantification of collagen area from Masson's Trichrome-stained kidney sections (*n* = 5/group, except for WT-NaHCO₃: *n* = 4), **g** BUN, **h** serum creatinine, **i** urine ACRs, **j** heatmaps showing renal mRNA expression of fibrotic markers (*Col1a1, Col1a2, Col3a1, Fn1, Acta2*), pro-inflammatory markers (*Tnfα, Il6, Ccl2, Ccl5, Il1β*), kidney injury markers (*Ngal, Kim1*), and pEMT markers (*Snai1, Zeb*) (values are shown in Supplementary Fig. 10). **c, d, g–i** WT-NaHCO₃ (*n* = 4), CKO-NaHCO₃ (*n* = 5), WT-FA (*n* = 8), CKO-FA (*n* = 7). **c, d, f–i** Data are shown as box-and-whisker with median (middle line), 25th–75th percentiles (box), and minimum–maximum values (whiskers); two-way ANOVA with Sidak's correction. Source data are provided as a Source data file.

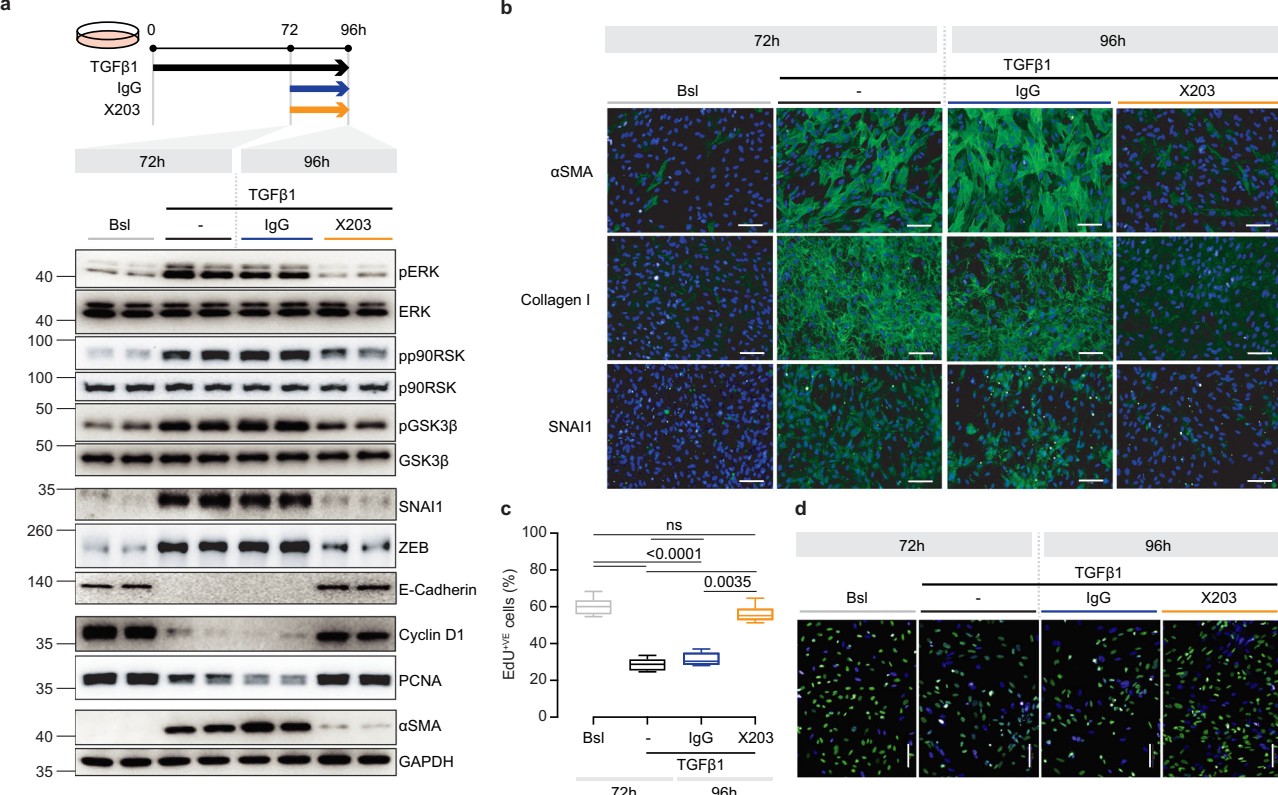

**Fig. 8 | Anti-IL11 reverses the partial epithelial-to-mesenchymal transition of human tubular epithelial cells. a** Schematic and Western blots (representative dataset from *n* = 4/group) of pERK, ERK, pp90RSK, p90RSK, pGSK3β, GSK3β, SNAI1, ZEB, E-Cadherin, PCNA, Cyclin D1, αSMA, and GAPDH for TEC reversal experiment: TECs were stimulated with TGFβ for 72 h followed by addition of IgG/X203 for another 24 h. **b** Representative IF images of αSMA[+ve] and SNAI1[+ve] cells and Collagen 1 immunostaining (scale bars: 100 μm, representative dataset from

*n* = 3/group), and **c** quantification (*n* = 14/group) and **d** representative IF images (scale bars: 100 μm, representative dataset from *n* = 3/group) of EdU[+ve] cells for TEC reversal experiments as shown in (**a**). **c** Data are shown as box-and-whisker with median (middle line), 25th–75th percentiles (box), and minimum–maximum values (whiskers); Kruskal–Wallis with Dunn's correction. Bsl: Baseline. Source data are provided as a Source data file.

accordance with the SingHealth Institutional Animal Care and Use Committee (IACUC). All mice were provided food and water ad libitum.

## Animal models

Mice were housed in temperatures of 21–24 °C with 40–70% humidity on a 12 h light/12 h dark cycle and provided with food and water ad libitum. For mouse model of folic acid (FA)-induced acute kidney injury (AKI), AKI was induced by intraperitoneal (IP) injection of folic acid (FA, 200 mg/kg) in vehicle (0.3 M NaHCO₃) to 10–12-week-old male mice; control mice were administered vehicle alone. Unilateral ureteral obstruction (UUO) surgeries were carried out on 12–13-week-old male mice to generate a mouse model accelerated chronic kidney disease. Briefly, mice were anesthetized by IP injection of

ketamine (100 mg/kg) /xylazine (10 mg/kg) and full depth of anesthesia was accessed with the pedal reflex. Mice were then shaved on the left side of the abdomen. A vertical incision was made through the skin with a scalpel, a second incision was made through the peritoneum to reveal the kidney. Using forceps, the left kidney was brought to the surface and the ureter was tied with surgical silk, twice, below the kidney. The ligated kidney was placed back into its correct anatomical position and sterile saline was added to replenish the loss of fluid. The incisions were then sutured. Animals were post-operatively treated with antibiotic enrofloxacin (15 mg/kg, SC) and analgesic buprenorphine (0.1 mg/kg, SC) for three consecutive days. Mice were sacrificed 9 days post UUO (D10). The number of mice used in each experiment is outlined in the respective figure legends

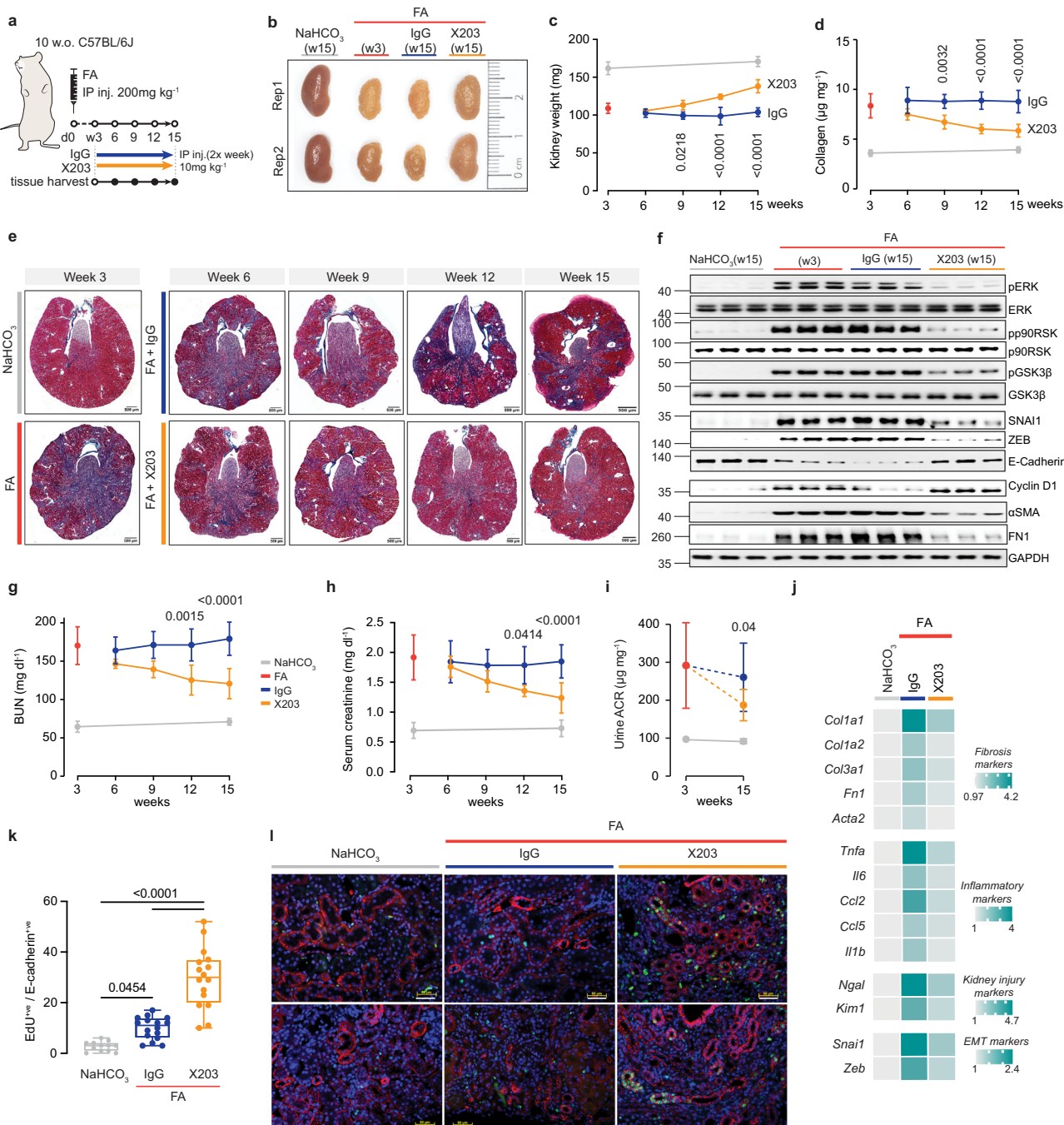

**Fig. 9 | Anti-IL11 promotes kidney regeneration and reverses renal dysfunction in chronic kidney disease. a** Schematic of X203 therapeutic dosing regimen for CKD reversal experiments shown in (**b–l**): X203 or IgG (IP, 10 mg/kg) was administered 2X/week for 12 weeks from day 21 (week 3) after FA injury, when CKD is established and sustained. Blood and kidneys were collected every 3 weeks; full course serum and kidney phenotyping study were performed at the study endpoint (week 15: 12 weeks after the start of X203 therapy); blood and kidneys from control mice i.e., those receiving IP injection of vehicle control (0.3 M NaHCO₃) were collected at week 3 and 15. **b** Representative kidney gross anatomy, **c** kidney weights (NaHCO₃ (W3), FA (W3) (n = 11/group), FA + IgG (W6, W9), FA + X203 (W6, W12) (n = 4/group), FA + X203 (W9), FA + IgG (W12), NaHCO₃ (W15) (n = 5/group), FA + IgG/X203 (W15) n = 13/group)), **d** renal collagen content by hydroxyproline assay, **e** representative Masson's Trichrome images of whole kidney cross section (scale bars: 500 μm, representative dataset from n = 4/group for kidneys collected at week 6, 9, 12 or from n = 5/group for week 3 and 15), **f** Western blots showing renal ERK and p90RSK activation, GSK3β inactivation, and SNAI1, ZEB, E-Cadherin, Cyclin D1, αSMA, FN1, GAPDH expression (representative dataset from

n = 5/group), **g** BUN, **h** serum creatinine, **i** urine ACR (NaHCO₃ (W3, W15) (n = 5/group), FA (W3) (n = 6), FA + IgG/X203 (W15) (n = 9/group). **d, g, h** NaHCO₃ (W3), FA (W3) (n = 11/group), FA + IgG (W6, W9), FA + X203 (W6) (n = 4/group), FA + X203 (W9, W12), FA + IgG (W12), NaHCO₃ (W15) (n = 5/group), FA + IgG/X203 (W15) (n = 13/group). **j** Heatmaps of renal mRNA expression of fibrosis, inflammation, kidney injury, and EMT markers (values are shown in Supplementary Fig. 12) from control, IgG- and X203-injected mice (week 15). **k** Representative immuno-fluorescence images of EdU (green) and E-Cadherin (red) expression and **l** quantification of EdU⁺ᵛᵉ E-Cadherin⁺ᵛᵉ cells in the kidneys of mice receiving either IgG or X203 reversal dosing regimen for 6 weeks starting from day 21 post FA administration (representative dataset from n = 3/group for NaHCO₃ and n = 4/group for FA + IgG and FA + X203 group; scale bars: 100 μm). **c, d, g–i** Data are shown as mean ± SD; **k** data are shown as box-and-whisker with median (middle line), 25th–75th percentiles (box), and minimum–maximum values (whiskers). **c, d, g, h** two-way ANOVA with Sidak's correction; **i** two-tailed Student's t-test; **k** one-way ANOVA with Tukey's correction. Source data are provided as a Source data file.

## In vivo administration of anti-IL11 (X203 or MAB218) or anti-TGFβ

10–13-week-old C57BL/6J male mice (InVivos, Singapore (SG)) were administered anti-IL11 (X203 or MAB218), anti-TGFβ (1D11), or IgG isotype control (11E10) by intraperitoneal injection at different times, doses (1–20 mg/kg), and durations depending on the experiments as outlined in the main text, figures, and/or figure legends.

## Il11-EGFP mice

Transgenic mice with *EGFP* constitutively knocked-in to the *Il11* gene were generated by Cyagen Biosciences Inc in C57BL/6J background[21]. For FA experiments, 10–12-week-old male $Il11^{EGFP/+}$ mice were IP injected with 150 mg/kg of FA in vehicle (0.3 M NaHCO$_3$), aged-matched $Il11^{EGFP/+}$ littermates receiving equal volume of vehicle were used as controls. Mice were sacrificed at different time points as outlined in the main text, figures, and/or figure legends. For UUO experiments, 12–13-week-old male $Il11^{EGFP/+}$ mice were subjected to UUO surgery as described above; contralateral (right) kidneys were used as controls.

## Tubular epithelial cells (TEC)-specific Il11ra1-deleted mice

To induce the specific deletion of *Il11ra1*, we crossed Ksp1.3/Cre transgenic mice (B6.Cg-Tg(Cdh16-cre)91Igr/J, Jackson Laboratory), which express Cre recombinase under the control of Cadherin 16 (*Cdh16*) promoter, with homozygous *Il11ra1*-floxed mice. Exons 4 to 7 of the *Il11ra1* gene were flanked by loxP sites ($Il11ra1^{loxP/loxP}$) allowing for deletion of *Il11ra1* upon Cre recombinase-mediated excision[21]. Knockdown efficiency was determined by Western blotting of renal IL11RA expression. 10–12-week-old male $Cdh16^{Cre/+}Il11ra1^{loxP/loxP}$ (CKO) and $Cdh16^{-/-}Il11ra1^{loxP/loxP}$ mice, which were used as wild-type (WT) controls, were injected with 200 mg/kg of FA or an equal amount of vehicle.

## Urine collection

Two days prior to the end of the study, 24-h urine collections were obtained from mice that are housed in individual metabolic cages with free access to water and food.

## EdU incorporation studies

12-week-old male C57BL/6J mice (InVivos, Singapore (SG)) were administered 20 mg/kg (2x/week) of anti-IL11 (X203) or IgG for 6 weeks, starting from day 21 post FA (200 mg/kg) injection. Mice were administered a daily intraperitoneal injection of EdU (20 mg/kg) in sterile, phosphate-buffered saline (PBS, pH 7.4) for 5 days before they were sacrificed at week 9 post FA.

## Cell culture

Cells were grown and maintained at 37 °C and 5% CO$_2$. The growth medium was renewed every 2–3 days and cells were passaged at 80–90% confluence using standard trypsinization techniques. All the experiments were carried out at P3. Cells were serum-starved for 16 h prior to stimulations. Stimulated cells were compared to unstimulated cells that have been grown for the same duration under the same conditions (serum-free media), but without the stimuli. Primary human renal proximal tubular epithelial cells (TECs, 4100, lot 19754, ScienCell) isolated from a healthy human kidney of 20-year-old female were grown and maintained in complete renal epithelial cell growth medium (4101, ScienCell) containing fetal bovine serum (FBS, 0010, ScienCell), epithelial cell growth supplement (EpiCGS, 4152, ScienCell) and antibiotic solution (P/S, 0503, ScienCell). Primary human kidney fibroblasts (P10666, Lot 20115ty, InnoProt) isolated from a healthy human kidney (59-year-old male) were maintained in a fibroblast medium-PLUS (P60108-PLUS, Innoprot).

## Operetta platform phenotyping assay and image analysis

Cells were seeded in 96-well CellCarrier plates (600550, PerkinElmer) at a density of $1 \times 10^4$ cells per well. Following experimental conditions, cells were fixed in 4% paraformaldehyde (PFA), permeabilized with 0.1% Triton X-100 in PBS. EdU–AlexaFluor488 was incorporated using a Click-iT EdU labeling kit (C10350, Life Technologies) according to the manufacturer's protocol. Non-specific binding sites were blocked in blocking solution (PBS containing 0.5% BSA and 0.1% Tween-20) for 1 h at RT. Cells were incubated overnight (4 °C) with primary antibodies (1:500 in blocking solution) i.e., αSMA, Collagen I, SNAI1 followed by incubation with the appropriate AlexaFluor488-conjugated secondary antibodies for 1 h (1:1000 in blocking solution, RT). Cells were counterstained with DAPI (1 μg/ml) in blocking solution. Plates were scanned and images were collected with an Operetta high-content imaging system (1483, PerkinElmer). Each condition was assayed from a minimum of seven fields per well. The quantification of αSMA$^{+ve}$, SNAI1$^{+ve}$, and Edu$^{+ve}$ cells was done using Harmony software version 3.5.2. and the percentage of activated fibroblasts/total cell number (αSMA$^{+ve}$) was determined for each field. The measurement of fluorescence intensity per area (normalized to the number of cells) of Collagen I was performed with Columbus 2.7.1.

## siRNA knockdown

TECs were transfected using Lipofectamine RNAiMax (Life Technologies) following the manufacturer's instructions for reverse transfection. In a 96-well plate, $6 \times 10^3$ cells were seeded per well and transfected with 25 nM On-Targetplus siRNAs (siSNAI1: L0010847-01-0005, siNT: D-001810-10-05, Dharmacon) in a medium consisting of serum-free Opti-MEM and complete TEC medium, combined in a 1:9 ratio. After 24 h of transfection, the medium was changed and TECs were serum-starved for 16 h prior to stimulation with IL11 or TGFβ.

## RNA sequencing

Primary human TECs were seeded in 6-well plates at a density of $2.5 \times 10^5$ cells/well. Following overnight starvation and stimulation with IL11 (5 ng/ml) for either 1, 6, or 24 h, total RNA was extracted with RNeasy column (Qiagen) purification according to the manufacturer's instruction to generate PolyA+ RNA sequencing libraries with the Truseq Stranded mRNA kit (Illumina). Barcoded RNA sequencing libraries were pooled and sequenced on the Illumina HiSeq 2500 platform using 75-bp paired-end sequencing chemistry. Raw sequencing data (.bcl files) were demultiplexed into fastq files with Illumina's bcl2fastq v2.16.0.10 based on unique index pairs. Adapters and low-quality reads were trimmed using Trimmomatic V0.36[53], retaining reads >20 nucleotides. Read mapping was carried out using STAR v2.5.2b[54] to the Ensembl Human GRCh38 v86 reference genome using standard parameters for full-length RNA-seq in the ENCODE project. Read counting was carried out using featureCounts[55] to get gene-level quantification of genomic features: featureCounts -t exon -g gene_id -O -s 2 -J -p -R -G. Quality check for sequencing data and mapping data was carried out using FastQC v0.11.5[56] and MultiQC[57]. Differential expression (DE) was performed with DESeq2 v1.14.1[58] by using raw reads count from featureCounts. Baseline samples were used as the reference level for each paired two-group comparison with 1 h, 6 h, and 24 h samples. Gene set enrichment analysis was carried out using fgsea R package, MSigDB Hallmark, and Gene ontology gene sets with 100,000 iterations[59,60]. The "stat" column of the DESeq2 result output was used to rank the genes as input for the enrichment analysis.

## Immunofluorescence (IF)

TECs were seeded on 8-well chamber slides ($1.5 \times 10^4$ cells/well) 24 h before the staining. Cells were fixed at 4% PFA for 20 minutes, washed with PBS, and non-specific sites were blocked with 5% BSA in PBS for 2 h. Cells were incubated with IL11RA antibody (1:200 in PBST (PBS containing 0.1% of Triton X-100)) or IgG isotype control (1:200 in PBST,

Aldevron) overnight (4 °C), followed by incubation with anti-rabbit Alexa Fluor 488 (1:500 in PBST) for 1 h. Chamber slides were dried in the dark and 5 drops of mounting medium with DAPI were added to the slides for 15 min prior to imaging by fluorescence microscope (Leica). Kidneys from *Il11*-EGFP mice and EdU-injected mice were rinsed in cold PBS, patted dry with a lint-free paper, and cryo-molded in OCT compound (4583, Tissue-Tek®). After the OCT compound was frozen, kidneys were wrapped in aluminum foil and stored in −80 °C. Cryo-embedded kidneys were cryosectioned (−20 °C) with a thickness of 7 μm and allowed to dry on the slides for 1 h (RT). Kidney sections were fixed in cold acetone for 15 min prior to brief PBS washes, permeabilized with 0.1% Triton X-100 (T8787, Sigma), and blocked with 2.5% normal goat serum (S-1012, Vector Labs) for 1 h (RT). Kidney sections were incubated with GFP (1:1000 in PBST) and E-Cadherin (1:1000 in PBST) primary antibodies overnight (4 °C), followed by incubation with the appropriate Alexa Fluor 488/647 secondary antibodies (1:250 in PBST) for 1 h (RT). For EdU detection, kidney sections were incubated with the reaction cocktail from Baseclick's EdU IV Imaging Kit 488 L (BCK488-IV-IM-L) according to the manufacturer's protocol. EdU and E-Cadherin double positive cells were quantified in blinded fashion from 4 randomly selected 200X field images using the cell counter function in ImageJ software. DAPI was used to stain the nuclei prior to imaging by fluorescence microscope (Leica).

## Western blot

Western blot was carried out on total protein extracts from TECs and kidney tissues. Cells and kidneys were lysed in radio-immunoprecipitation assay (RIPA) buffer containing protease and phosphatase inhibitors (Thermo Scientifics), followed by centrifugation to clear the lysate. Protein concentrations were determined by Bradford assay (Bio-Rad). Protein lysates were separated by SDS-PAGE, transferred to PVDF membrane, and subjected to immunoblot analysis for various antibodies as outlined in the main text, figures, or and/or figure legends. All primary antibodies were used at 1:1000 dilution in TBST (TBS with 0.2% of Triton X-100) and secondary antibodies were diluted 1:2000 in TBST containing 3% BSA. Proteins were visualized using the ECL detection system (Pierce) with the appropriate secondary antibodies: anti-rabbit HRP or anti-mouse HRP. Densitometry analyses are provided in Source data file 2. Uncropped blots for the main and supplementary figures are provided in Source data file 3 and at the end of supplementary information file, respectively.

## Quantitative polymerase chain reaction (qPCR)

Total RNA was extracted from snap-frozen kidney tissues using Trizol (Invitrogen) followed by RNeasy column (Qiagen) purification. cDNAs were synthesized with iScript™ cDNA synthesis kit (Bio-Rad) according to manufacturer's instructions. Gene expression analysis was performed on duplicate samples with either TaqMan (Applied Biosystems) or fast SYBR green (Qiagen) technology using StepOnePlus™ (Applied Biosystem) over 40 cycles. Expression data were normalized to *GAPDH* mRNA expression and fold change was calculated using $2^{-\Delta\Delta Ct}$ method. The SYBR primer sequences are listed in Supplementary Table 1. TaqMan probes used in this study are as follow: Mm01256744_ml (Mouse *Fn1*), Mm00446190_m1 (Mouse *Il6*), Mm00434162_m1 (Mouse *Il11*), Hs02786624_g1 (Human *GAPDH*).

## Colorimetric assays and ELISA

The levels of IL11 in equal volumes of cell supernatant were measured using Human IL11 Quantikine ELISA kit (D1100, R&D Systems). Total hydroxyproline content in mouse kidneys was measured using Quickzyme Total Collagen assay kit (QZBtotco15, Quickzyme Biosciences). The levels of blood urea nitrogen (BUN) in mouse serum were measured using Urea Assay Kit (ab83362, Abcam). Albumin and creatinine levels were measured using Mouse Albumin ELISA kit (ab108792, Abcam) and Creatinine Assay Kit (ab65340, Abcam), respectively. All

ELISA and colorimetric assays were performed according to the manufacturer's protocol.

## Histology

For Masson's Trichrome's staining, kidney tissues were fixed for 48 h at RT in 10% neutral-buffered formalin (NBF), dehydrated, embedded in paraffin blocks, and sectioned at 4 μm. Sections were then stained with Masson's Trichrome according to the standard protocol and examined by light microscopy. For immunohistochemistry, kidneys were processed as mentioned above. Following dewaxing and antigen retrieval processes (98 °C for 20 min in citrate buffer), kidney sections were permeabilized with 0.1% Triton X-100 (Sigma), incubated with 3% $H_2O_2$ (Sigma), and blocked with 2.5% normal horse serum (S-2012, Vector Labs). Kidney sections were incubated with primary antibody (X203 and X209 at 1:250 dilution in PBST) overnight (4 °C) and visualized using an ImmPRESS HRP horse anti-mouse IgG polymer detection kit (MP-7402, Vector Labs) with ImmPACT DAB Peroxidase Substrate (SK-4105, Vector Labs). Hematoxylin (H-3401, Vector Labs) was used to stain the nuclei prior to imaging by light microscopy.

## Statistical analysis

Statistical analyses were performed using GraphPad Prism software (version 9.4.1). Datasets were tested for normality with Shapiro–Wilk tests and outlier tests were performed using the ROUT method. For normally distributed data, statistical significance between control and experimental groups were analyzed by two-sided Student's t tests or by one-way ANOVA as indicated in the figure legends. *P* values were corrected for multiple testing according to Dunnett's (when several experimental groups were compared to a single control group) or Tukey (when several conditions were compared to each other within one experiment). Non-parametric test (Kruskal–Wallis with Dunn's correction in place of ANOVA) was conducted for non-normally distributed data. Comparison analysis for two parameters from two different groups were performed by two-way ANOVA and corrected with Sidak's multiple comparisons when the means were compared to each other. The criterion for statistical significance was $P < 0.05$. The complete list of exact *p* values is provided in Supplementary Data 2.

## Reporting summary

Further information on research design is available in the Nature Portfolio Reporting Summary linked to this article.

## Data availability

All data are available within the Article or Supplementary Information. The RNA-seq data reported in this study have been deposited in the GEO database under the NCBI GEO accession number: GSE199080. Raw data are provided in Source data file 1, densitometry analysis is provided in Source data file 2, and uncropped blots for main figures are provided in Source data file 3. Source data are provided with this paper.

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

## Acknowledgements

This research is supported by NMRC/OFYIRG/0053/2017 (A.A.W.), NMRC MOH-OFIRG21nov-0006 (A.A.W.), Duke-NUS KBrFA/2022/0057 from Khoo Foundation (A.A.W.), NMRC/STaR/0029/2017 (S.A.C.), NMRC MOH-CIRG18nov-0002 (S.A.C.), Goh Foundation (S.A.C.), Tanoto Foundation (S.A.C.), Leducq Foundation 16CVD03 (S.A.C. and N.H.), ERC advanced grant under the European Union Horizon 2020 Research and Innovation Program (AdG788970) (N.H.).

## Author contributions

A.A.W. and S.A.C. conceived, designed, and funded the study. A.A.W., S.V., S.G.S., J.T., J.W.T.G., and H.M.C., performed in vitro cell culture, in vivo studies, biochemistry, and molecular biology experiments. S.G.S. and S.Y.L. performed histology analysis. C.M.B.K., J.H., E.P., N.H., and T.M.C. provided resources. A.A.W., S.S., and S.A.C. analyzed the data. A.A.W., E.A. W.W.L., T.M.C., and S.A.C., prepared the manuscript with input from co-authors.

## Competing interests

A.A.W., S.S., and S.A.C. are co-inventors of the US Patent 11339216 (Treatment of kidney injury). S.S and S.A.C are co-inventors of the published patents: WO/2017/103108 (TREATMENT OF FIBROSIS), WO/2018/109174 (IL11 ANTIBODIES), WO/2018/109170 (IL11RA ANTIBODIES). S.S. and S.A.C. are co-founders and shareholders of Enleofen Bio PTE LTD, a company that made anti-IL11 therapeutics, which were acquired for further development by Boehringer Ingelheim in 2019. The remaining authors declare no competing interests.
