## [Peer Review File · Nature Communications]

Reviewers' Comments:

Reviewer #1:

Remarks to the Author:

The investigators report that IL11 is upregulated with kidney injury and the IL11 null mouse as well as mice treated with neutralizing IL11 antibody have attenuated partial EMT (pEMT), fibrosis and kidney dysfunction. TGF β induces autocrine IL11/ERK-dependent pEMT leading to paracrine, IL11-mediated fibroblast activation. Mice with tubule specific deletion of IL11 RA1 are protected from pEMT, inflammation, fibrosis, and renal failure. In a mouse model of chronic kidney disease, administration of anti-IL11 reverses fibrosis, and, the authors conclude, regenerates parenchyma, and restores renal function. The paper is interesting and important, but there are a number of issues that the authors should address.

1. Figure 1F: the data presented relate to the IL11 knockout mouse. Is the IL11 receptor normally expressed or is it modulated in the IL11 knockout? The authors suggest that the α SMA upregulation in Figure 1F in the IL11 knockout is expected since α SMA is "typically expressed in TECs undergoing pEMT and in myofibroblasts". There is some controversy about this concept. There are more data that suggest that the TECs do not express α SMA even with pEMT, so that this is likely to be coming from the myofibroblast. α SMA is expressed with pEMT in vitro, but many do not find this in vivo.
2. The data in figure 6 are impressive showing increase of kidney weight and the decrease of collagen production over time of treatment with X203. Functionally, in Figure 6K and L, there are decreases in BUN, ACR and serum creatinine from 3 to 16 weeks in the X203 treatment cases. In Figure 6K, L and M, it would be useful to separate out the baseline values of the two treatment groups. As it is presented, there is one baseline value presented for both groups and another for the sham. Separation out will assure the reader that there were no systematic differences in the IgG control group vs the X203 group at 3 weeks prior to treatment.
3. In the supplemental data under animal models, I am not sure I would characterize the UUO model as a surgically induced acute kidney injury model. Since the ureteral tie is not removed, this is really a model of accelerated chronic kidney disease as reflected by the extensive changes on histology.
4. Although the manuscript is interesting and important, similar findings have been observed and published in cardiovascular fibrosis, lung fibrosis, and liver fibrosis, reducing a bit the novelty of the finding.
5. A great deal in the paper depends upon the specificity of the reagent used to block IL11. In looking back to some of the papers referred to when the authors first introduce this antibody in the manuscript, it is difficult to find out anything about the specificity of the antibody. It would be useful to include enough information to enable the reader to be convinced of its specificity as so much of the interpretation of the results depends on these.
6. In figure 1A, there is an expression of E-Cadherin in the kidney which appears to be reduced somewhat with folic acid. The authors should present the N-Cadherin staining. Generally E-Cadherin is more commonly expressed in the distal nephron, whereas N-Cadherin is expressed in the proximal nephron.

Reviewer #2:

Remarks to the Author:

This is an interesting manuscript that investigates the role of interleukin 11 (IL11) in tubular epithelial cell (TEC) partial EMT and kidney repair/fibrosis. Using in vivo and in vitro as well as conditional knockout (cKO) approaches, the authors demonstrate that IL11 stimulates partial EMT in damaged TECs after kidney injury, triggering an ERK/P90RSK/GSK3 β /SNAI1 signal cascade leading to impaired renal function. Blockade of IL11 by neutralizing antibody or cKO of IL11 receptor ameliorated kidney fibrosis and restored renal function. They conclude that

therapeutically inhibition of IL11 signaling promotes kidney repair and attenuates renal fibrosis by hampering EMT.

This study is an extension of previous studies from the same group showing profibrotic effect of IL11 after kidney and cardiac injury. While their previous work focused on interstitial fibroblasts, the present work investigated the role of IL11 in TECs. There are some concerns about the novelty of the study. The study is comprehensive and well performed, and for the most part, the experimental data are well presented and convincing. Several concerns need to be addressed.

Major comments:

1. Earlier work from the same group (Ref. 14) already reported that IL11 plays an important role in kidney fibrosis after injury, although the previous work focused on interstitial fibrosis. This not only impacts the novelty of the study, but also affects the interpretation of the results.
2. Blockade of IL11 signaling by neutralizing antibody will affect its action on all kinds of cells, including TEC and fibroblasts. Previous studies from the same group showed the importance of autocrine IL11 signaling in fibroblasts in kidney fibrosis, here they IL11 signaling on TEC almost completely blocked fibrotic lesions. How to reconcile these findings? What is the relevant importance of IL11 signaling on fibroblasts versus TEC?
3. Authors claimed the importance of ERK/P90RSK/GSK3 β /SNAL1 axis in kidney fibrosis (Figure 4F). It is well known that GSK3 β controls many substrates; in fact, one major downstream target is beta-catenin, which also plays an important role in TEC partial EMT. Therefore, it should be explored whether IL11 can activate β -catenin signaling and cause TEC pEMT.
4. Figure 2. Blockade of IL11 by X203 inhibited pEMT and induced Cyclin D1. Authors claimed X203 induces TEC proliferation/regeneration. However, no localization of cyclin D1 is provided. Similarly, staining for ERK, P90RSK and SNAL1 needs to be provided to show this signal cascade took place in renal tubules.
5. Figure 5. The cKO mice was generated by crossing Cadh16-Cre mice and IL11R-floxed mice. Information on the characterization of these mice is not given. Which tubular segment express Cadh16? How efficient the cKO is? Staining and confirmation of cKO needs to be provided.
6. It would strengthen the studies if the expression of IL-11 in the clinical specimens from CKD patients can be provided, such as blood, urine, or kidney tissue.

Other comments:

7. Figure 2C and Figure 6K only show the results of BUN, please add the results of serum creatinine.
8. Please add the quantitative results for Masson staining (Figures 2-3, Figures 5-6, and Supplementary Figure 1-4), and each group is at least 5 mice.
9. The Western blot results of animal experiments throughout the manuscript only show representative images (n=3). Please add the quantitative and statistical results of Western blots for all animals.
10. Figure 4A should add a group of IL11+X203.

Reviewer #3:

Remarks to the Author:

IL11 is known to regulate cell fibrosis in many cell types and tissues (and there is some indicative data in kidney also from this group) also EMT in some cells, in particular cancer cells. This study therefore adds some information in the kidney. The study also indicates that IL11 may be targeted

to regenerate kidneys after injury has occurred although the authors require further experiments to prove this. The mechanisms by which this occurs has not been thoroughly explored. It would be useful to determine changes in the transcriptome and more specifically it would be superior to investigate this at a single cell level rather than to investigate some factors associated with fibrosis, EMT and inflammation. The study is a great starting point but requires further investigation to determine the mechanisms by which IL11 acts specifically in the diseased/injured kidney.

There are some specific points the authors may also consider.

Results:

There are too many abbreviations that make the paper difficult to follow – please reduce some such as FA and TECs.

Figure 1:

Fig 1A. Need to complete statistical analysis to prove signal persists rather than just showing a representative IF image. Can't say something is less common unless you prove it. The samples size is small with N=3 and parametric tests will not be able to be completed as data normality will not be able to be assessed with n=3.

Fig 1B. IF is notoriously difficult to quantitate and again this needs to be completed to make statements such as 'very high levels'. As it stands you can say it localises to the cells. What concentrations of antibodies and IgG controls were used. I ask this as IL11Ra has been notoriously difficult to stain for using the commercial antibodies used in this study.

TGF beta is well known to stimulate the expression and secretion of IL11 in many cell types. Whether there is an autocrine or paracrine effect is not completely proven and it could equally be either.

Fig 1D. Phosphorylation (p) often occurs within minutes if it is a direct effect after IL11 stimulation acting via its specific receptor, in particular in cells in culture that contain IL11Ra. The p of ERK is apparent only after 8 hours. This suggests that IL11 may signal via another pathway initially, perhaps the JAK/STAT pathway which it is well known to signal via, and primarily signal via in many cell types. There is also no quantitation of the Westerns. Please show the densitometry and statistical analysis. I just noted you have an N=1. Is this correct? Please repeat to get the required N's as determined by power analysis to ensure statistical significance can be determined.

Fig 1E. The authors used one dose of FA 200mg/kg and then samples taken at Day 28. Why was this dose chosen and why were they left for 28 days before samples were collected? Is this a standard protocol? This is a very long time to determine whether the effects noted 28 days later are caused by loss of IL11 signaling directly or were associated with IL11 signalling indirectly. Is it also that the effect could be as a cause of the disease that is presumably apparent after 28 days.

Fig 1F, please show densitometric analysis and statistical tests – likely non-parametric tests as you cannot test for normality with an N=3.

Fig 1G. Not sure what G is showing precisely (from the legend). Please clarify.

Not sure what this refers to:

(C-D) TGFβ1/IL11 (5 ng/ml); 24 hours.

Figure 2. The authors need to demonstrate / comment on the half-life of the antibodies and determine if they reach the site of interest by demonstrating blocking of specific downstream signalling, such as pSTAT3 or ERK, after perhaps a single injection and collection of tissue of interest ie. Kidney over a short time frame. Could the effects could be off target effects? There are no good effective IL11 signalling blocking antibodies as far as I can determine from the literature. This is again the same for Figs 3 and 4.

I am not quite sure how the neutralisation of TGF beta experiment adds to the study aims. Did the authors hope to determine the specific mechanistic differences between the two agents in the kidney. The authors need to show that the neutralising antibodies neutralise their specific actions

in the kidney, or are they effects secondary/indirect.

The same quantitation if the Westerns is also required.

I am not sure how the statistical analysis was completed for the immunofluorescence as in Suppl Figure 6F. There appears to be substantial overlap for Collagen 1 for IL11 DMSO and IL11- and there is a statistical difference between the two. The representative images in Suppl Fig 6E also appear to be very similar. It would be ideal to see the details for the quantitation and statistical analysis. Again, same with the Western blots.

Figure 4G (and Suppl Fig 8) – error in ID10 – Should it read ID11?

Figure 6 – Time frame for treatment with anti-IL11 is 24 hours. It is difficult to tell whether the effects seen are due specifically by IL11 or secondary effects. Have the authors trialled treatment with anti-IL11 for shorter time periods and completed a time course in vivo and investigated the effects on the JAK/STAT pathway (specifically activation) which IL11 primarily signals via. This may indicate direct and/or indirect effects of IL11 and also determine whether the neutralising antibody reaches the site of interest and how long it is present at the site of interest to work.

The discussion has some repetition and is confusing to read and often repeats the results. Can it be re-written/organised to start with for instance - the main findings of the study rather than what is already known about kidney injury and TECs which should be in the introduction. The specific results do not need to be repeated in the discussion section.

Point-by-point responses to the comments made by Reviewers at Nature Communications

Reviewer #1 (Remarks to the Author):

The investigators report that IL11 is upregulated with kidney injury and the IL11 null mouse as well as mice treated with neutralizing IL11 antibody have attenuated partial EMT (pEMT), fibrosis and kidney dysfunction. TGF β induces autocrine IL11/ERK-dependent pEMT leading to paracrine, IL11-mediated fibroblast activation. Mice with tubule specific deletion of IL11RA1 are protected from pEMT, inflammation, fibrosis, and renal failure. In a mouse model of chronic kidney disease, administration of anti-IL11 reverses fibrosis, and, the authors conclude, regenerates parenchyma, and restores renal function. The paper is interesting and important, but there are a number of issues that the authors should address.

Author's response: We thank the Reviewer for his/her supportive comments and constructive suggestions.

1. Figure 1F: the data presented relate to the IL11 knockout mouse. Is the IL11 receptor normally expressed or is it modulated in the IL11 knockout?

Author's response: The *Il11* KO strategy targets the *Il11* locus and it is not clear how this would modify *Il11ral* expression or, even if it did, how this would signal in the absence of IL11 itself. However, we have looked into this for the Reviewer and found that IL11RA1 levels are normal in the IL11 KO at baseline by immunoblotting and IHC (**Rebuttal Fig. 1**). We have also added these data as part of **Fig. 2B** and **Supplementary Fig. 2B** in the revised manuscript. Following kidney injury, IL11RA1 levels are increased in WT mice but not in *Il11* KO, which was also seen in mice with TEC-specific *Il11ral* deletion (**Fig. 7B**; **Supplementary Fig. 10A** and also shown in **Rebuttal Fig. 6**). This infers IL11-induced IL11RA1 expression in TECs in the injured kidney, which was apparent on immunohistochemistry analysis (**Rebuttal Fig. 1**).

Rebuttal Figure 1. IL11RA1 expression in the kidneys of *Il11* KO mice (A) Western blots (representative dataset from n=5/group) and (B) densitometry analysis (n=5/group) of WT and *Il11*^{-/-} (KO) mice administered with folic acid (FA) or vehicle (NAHCO₃) showing renal expression of IL11RA1 and GAPDH. (C) Immunohistochemistry images of kidneys probed for IL11RA1 (X209; 1:250 in 0.1% PBST) (scale bars: 100 μm; representative dataset from n=3/group). (B) Data are shown as mean±SD, one-way ANOVA with Sidak comparisons.

The authors suggest that the αSMA upregulation in Figure 1F in the *IL11* knockout is expected since αSMA is “typically expressed in TECs undergoing pEMT and in myofibroblasts”. There is some controversy about this concept. There are more data that suggest that the TECs do not express αSMA even with pEMT, so that this is likely to be coming from the myofibroblast. αSMA is expressed with pEMT in vitro, but many do not find this in vivo.

Author’s response: We agree with the reviewer that there is controversy in the literature on this subject, notably with respect to epithelial mesenchymal transition (EMT) [in full] as opposed to partial EMT (pEMT) that is also termed ‘failed repair’ by some. In our current study, we show IL11-stimulated upregulation of αSMA (and SNAI1, ZEB, Collagen) and concomitant downregulation of E-cadherin and proliferation markers (Cyclin D1) in primary cultures of human TECs. We also show TGFβ1-induced expression of αSMA, SNAI1, ZEB and collagen in primary human TECs is reversible using anti-IL11. We show concordant effects *in vivo* in mouse kidneys with TEC-specific deletion of *Il11ral*. Thus, while there is controversy relating to αSMA expression in TECs undergoing EMT (and the extent of this process), we show in our TEC systems that a mesenchymal program (including αSMA) is IL11 dependent. This fits with our recent finding that an IL11/ERK/LKB1 pathway is

important for mesenchymal programs across cell types¹. In our revision we more fully acknowledge the controversies.

2. The data in figure 6 are impressive showing increase of kidney weight and the decrease of collagen production over time of treatment with X203. Functionally, in Figure 6K and L, there are decreases in BUN, ACR and serum creatinine from 3 to 16 weeks in the X203 treatment cases. In Figure 6K, L and M, it would be useful to separate out the baseline values of the two treatment groups. As it is presented, there is one baseline value presented for both groups and another for the sham. Separation out will assure the reader that there were no systematic differences in the IgG control group vs the X203 group at 3 weeks prior to treatment.

Author's response: Three weeks after AKI mice were randomly assigned to (1) be sacrificed to establish baseline post-injury phenotypes or (2) receive either IgG or anti-IL11 for up to 12 weeks. As such, post-injury 'baseline' values of kidney weight, BUN, Cr, etc were established in a single group of mice that had to be culled to establish these values (e.g. kidney weight). The remaining mice, randomly assigned to receive either IgG or X203, were culled in serial cohorts (n=4/group) every 3 weeks to establish temporal changes in kidney weights/fibrosis, BUN/Cr, histology and signalling (up to week 15). After 3 weeks of therapy (week 6 of the experiment), mice randomly assigned to either IgG or X203 had indistinguishable renal phenotypes that were also unchanged from baseline post-injury phenotypes (week 3 of the experiment). Only at later time points was there linear and progressive change in renal physiology and restoration of kidney mass. For these reasons these data are presented as they were in the original submission and cannot be represented otherwise but show no difference in the treatment groups early on during the experiment.

3. In the supplemental data under animal models, I am not sure I would characterize the UUO model as a surgically induced acute kidney injury model. Since the ureteral tie is not removed, this is really a model of accelerated chronic kidney disease as reflected by the extensive changes on histology.

Author's response: We thank the Reviewer for this suggestion and have amended the text accordingly.

4. Although the manuscript is interesting and important, similar findings have been observed and published in cardiovascular fibrosis, lung fibrosis, and liver fibrosis, reducing a bit the novelty of the finding.

Author's response: We would like to point out that this manuscript is not a study of fibrosis but instead describes the primacy of IL11 signalling in TECs for kidney pathology, EMT and arrested regeneration, which precedes fibrosis and inflammation. We are not aware of any study to date showing therapeutic reversal of murine CKD combined with renal regeneration through regression of TEC pEMT and restoration of TEC proliferation (**Figs. 8 and 9; Supplementary Fig. 12**). A specific role for IL11 in TECs is not described; here we show this both *in vitro* and *in vivo*. The mechanism of IL11-stimulated ERK/p90RSK-mediated inactivation of GSK3 β leading to SNAIL1 upregulation has also not been described before.

Furthermore, in the revision we now present additional novel data on IL11-related ERK/STAT3 cross-talk, mediated via DUSP5, and define the TEC transcriptome over a time course of IL11 stimulation. This new experiment, stimulated by a comment from Reviewer 3, provides substantial new insights into the molecular processes activated in TECs following IL11 stimulation (notably pro-inflammatory and pro-mesenchymal effects). We also show new and novel data on TEC proliferation following anti-IL11 in the mouse model of CKD (**Fig. 9K-L**).

5. A great deal in the paper depends upon the specificity of the reagent used to block IL11. In looking back to some of the papers referred to when the authors first introduce this antibody in the manuscript, it is difficult to find out anything about the specificity of the antibody. It would be useful to include enough information to enable the reader to be convinced of its specificity as so much of the interpretation of the results depends on these.

Author's response: We can assure the Reviewer of the specificity of this particular antibody (X203) that, following humanisation, is now in development for human clinical trials. We cite papers showing that this antibody binds mouse IL11 with high affinity² and that the effects of X203 also phenocopy those of IL11 KO, IL11RA1 KO as well as anti-IL11RA antibodies used in other disease models, and in the current manuscript²⁻⁴. Furthermore, in the current manuscript we show that X203 has the same dose dependent molecular and physiological effects as a commercial anti-IL11 (MAB218) (**Fig. 3D and E; Supplementary Fig. 4**), which binds to a separate epitope⁵. We have gone even further to assure the reviewer and validate, once again, the specificity of the antibody using both western blotting and IHC in wildtype and IL11 KO mice in kidney tissues used in the current manuscript (**Rebuttal Fig. 2**; data are also shown as **Fig. 2B and Supplementary Fig. 2B**).

Rebuttal Figure 2. Validation of X203 specificity to IL11. (A) Western blots (representative dataset from n=5/group) and (B) densitometry analysis (n=5/group) of renal expression of IL11 and GAPDH (same GAPDH blot as Rebuttal Fig. 1) from WT and *Il11*^{-/-} (*Il11* KO) mice administered with folic acid (FA) or vehicle (NAHCO₃). (C) Immunohistochemistry images of kidneys probed for IL11 (X203; 1:250 in 0.1% PBST); staining images for IgG isotype control antibody (11E10; 1:250) are provided as negative controls (scale bars: 100 μm; representative dataset from n=3/group). (B) Data are shown as mean±SD, 2-way ANOVA with Sidak's comparisons.

6. In figure 1A, there is an expression of E-Cadherin in the kidney which appears to be reduced somewhat with folic acid. The authors should present the N-Cadherin staining. Generally E-Cadherin is more commonly expressed in the distal nephron, whereas N-Cadherin is expressed in the proximal nephron.

Author's response: E-Cadherin is an accepted marker of TEC polarity and known to be downregulated by SNAI1 in injured TECs as they dedifferentiate and become dysfunctional. We show IL11-dependent E-Cadherin downregulation and SNAI1 upregulation *in vivo* across models of kidney disease and also in TECs stimulated with a range of pathogenic factors *in vitro*. We are happy to provide additional staining on N-Cadherin for the reviewer (below). As this does not appear to add information over and above what we already show in the manuscript this is shown here for the reviewer's information only.

Rebuttal Figure 3. EGFP and N-Cadherin expression in UUO and FA-injured kidneys. Representative immunofluorescence (IF) images of EGFP and N-Cadherin expression in the kidneys of *Il1l1*^{EGFP/+} mice following (A) UUO or (B) folic acid injury of *Il1l1*^{EGFP/+} mice (representative dataset from n=3/group; scale bars: 100 μ m).

Reviewer #2 (Remarks to the Author):

This is an interesting manuscript that investigates the role of interleukin 11 (IL11) in tubular epithelial cell (TEC) partial EMT and kidney repair/fibrosis. Using in vivo and in vitro as well as conditional knockout (cKO) approaches, the authors demonstrate that IL11 stimulates partial EMT in damaged TECs after kidney injury, triggering an ERK/P90RSK/GSK3 β /SNAIL signal cascade leading to impaired renal function. Blockade of IL11 by neutralizing antibody or cKO of IL11 receptor ameliorated kidney fibrosis and restored renal function. They conclude that therapeutically inhibition of IL11 signaling promotes kidney repair and attenuates renal fibrosis by hampering EMT.

This study is an extension of previous studies from the same group showing profibrotic effect of IL11 after kidney and cardiac injury. While their previous work focused on interstitial fibroblasts, the present work investigated the role of IL11 in TECs. There are some concerns about the novelty of the study. The study is comprehensive and well performed, and for the most part, the experimental data are well presented and convincing. Several concerns need to be addressed.

Author's response: We thank the reviewer for his/her comments.

Major comments:

1. Earlier work from the same group (Ref. 14) already reported that IL11 plays an important role in kidney fibrosis after injury, although the previous work focused on interstitial fibrosis. This not only impacts the novelty of the study, but also affects the interpretation of the results.

Author's response: Previous work on IL11 in the kidney has focused on fibroblasts (and fibrosis) in acute or genetic models of kidney injury. We are not aware of any publication, to date, showing therapeutic reversal of renal dysfunction combined with renal regeneration in a mouse model of CKD, which we show. Furthermore, a role for IL11 in TEC dysfunction is not described, which we show here using a new TEC-specific *Il11ra1* knockout mouse. The mechanism of IL11-stimulated ERK/p90RSK-mediated inactivation of GSK3 β leading to SNAIL upregulation in TECs has not been identified before. In the revision, we provide new data relating to a time course of IL11-induced transcriptional changes in TECs (by RNA-seq) that reveals a novel axis of IL11-related STAT3-ERK cross-talk, regulated by DUSP5. We also now document, using EDU and E-cadherin counterstaining, that anti-IL11 therapy is specifically permissive for TEC regeneration *in vivo*. These data are novel and distinct.

2. Blockade of IL11 signaling by neutralizing antibody will affect its action on all kinds of cells, including TEC and fibroblasts. Previous studies from the same group showed the importance of autocrine IL11 signaling in fibroblasts in kidney fibrosis, here they IL11 signaling on TEC almost completely blocked fibrotic lesions. How to reconcile these findings? What is the relevant importance of IL11 signaling on fibroblasts versus TEC?

Author's response: The Reviewer refers to the crux of one of the major novel discoveries in our study. The initiating injury in many/most kidney diseases is in the epithelium (tubules and/or glomerulus) and signals from the damaged epithelium secondarily activate stromal pathology (fibrosis and inflammation) leading eventually to renal dysfunction. It is the case that anti-IL11 therapy will impact pathology in both the epithelium and the stroma, which is

an important point and likely underlies the large effect of anti-IL11 therapy in CKD that includes fibrosis reversal.

Here we show, using a novel model of TEC-specific *Il11ra1* deletion, that TEC-driven IL11-dependent effects are a critical initiating and propagating factor for kidney injury. The relative contributions of TEC vs fibroblast (or other cells) to the signalling events are apparent in studies of this model (**Fig. 7**) and are largely TEC-related and it is TEC proliferation (new data) that underlies the kidney regeneration seen with anti-IL11 in the CKD model. This axis of disease pathology is increasingly gaining attention and the subject of an entire sessions at recent meetings (e.g. “Epithelial Injury, Repair and Fibrosis”; at the Keystone “Tissue Fibrosis and Repair: Mechanisms, Human Disease and Therapies” meeting, July 2022).

3. Authors claimed the importance of ERK/P90RSK/GSK3 β /SNAIL axis in kidney fibrosis (Figure 4F). It is well known that GSK3 β controls many substrates; in fact, one major downstream target is beta-catenin, which also plays an important role in TEC partial EMT. Therefore, it should be explored whether IL11 can activate β -catenin signaling and cause TEC pEMT.

Author’s response: It is true that kinases have many substrates and that no single substrate (or indeed single kinase) will underlie a complex pathology such as AKI or CKD. However, it is established that GSK3 β is a critical determinant of SNAI1 expression^{6,7} and SNAI1 upregulation underlies TEC dysfunction and renal failure^{8,9}. For these specific reasons, we focused our signalling studies on defining the novel IL11/ERK/p90RSK/GSK3 β /SNAI1 axis. We are happy to also examine beta-catenin for the Reviewer and IL11 appears to have a bi-modal effect on its expression: initially downregulating it and then increasing its levels (**Rebuttal Fig. 4**), which would be expected to increase its signalling effects. While of some interest we believe this observation is tangential to the [already large] data presented in the manuscript and show it here only, for the reviewer.

Rebuttal Figure 4. β -catenin expression in TECs following IL11 stimulation. (A) Western blots (representative dataset from $n=4$ /group) and (B) densitometry analysis ($n=4$ /group) of β -Catenin and GAPDH in IL11-stimulated TECs over a time course. (B) Data are shown as mean \pm SD, one-way ANOVA with Dunnett's comparisons.

4. Figure 2. Blockade of IL11 by X203 inhibited pEMT and induced Cyclin D1. Authors claimed X203 induces TEC proliferation/regeneration. However, no localization of cyclin D1 is provided. Similarly, staining for ERK, P90RSK and SNAIL needs to be provided to show this signal cascade took place in renal tubules.

Author's response: To solve cell-type specific effects, we present extensive data on TECs *in vitro* and from kidneys of mice with TEC-specific deletion of *Il1ral* *in vivo*. To summarise the data: in primary cultures of human TECs, we showed IL11-dependent activation of ERK/p90RSK, upregulation of SNAI1 and downregulation of Cyclin D1 (and E-Cadherin), establishing this signalling axis in human TECs. Furthermore, in mice with kidney injury and TEC-specific deletion of *Il1ral*, we show inhibition of ERK/p90RSK activation, downregulation of SNAI1 and upregulation of Cyclin D1 (and E-Cadherin). This shows, using both direct assessment of IL11 gain-of-function in TECs *in vitro* and genetic loss-of-function of IL11 signalling *in vivo*, that these signalling events occur in TECs.

To more fully address the effect of IL11 on TEC regeneration *in vivo* we have performed a new set of experiments, whereby we administered EDU to mice with CKD at the same time as treating mice with either IgG or X203. This shows, we think quite beautifully, that anti-IL11 is permissive for TEC regeneration in the kidney tubules, which we also quantified in a blinded fashion (**Rebuttal Fig. 5** and also shown in **Fig. 9K-L**). We believe this is the first demonstration of therapeutically enabled TEC regeneration and reversal of CKD in a mouse model.

Rebuttal Figure 5. Administration of EDU to mice 6 weeks after antibody therapy was started. (A) Representative immunofluorescence images of EdU (green) and E-Cadherin (red) expression and (B) quantification of EdU⁺ve E-Cadherin⁺ve cells in the kidneys of mice receiving either IgG or X203 reversal dosing regimen for 6 weeks starting from week 3 post folic acid (FA) administration (representative dataset from n=3/group for control (NaHCO₃) group and n=4/group from FA+IgG and FA+X203 group; scale bars: 100 μm). Mice were sacrificed 9 weeks post FA-induced AKI.

5. Figure 5. The cKO mice was generated by crossing *Cadh16-Cre* mice and *IL11R*-floxed mice. Information on the characterization of these mice is not given. Which tubular segment express *Cadh16*? How efficient the cKO is? Staining and confirmation of cKO needs to be provided.

Author's response: The *Cadh16-Cre* mice is an established line that has been used in the literature to delete genes in TECs. To quote a paper from Nature Medicine from 2015 “Ksp-cadherin (cadherin-16, encoded by *Cadh16*) is a kidney-specific cadherin that is expressed in renal epithelial cells both in the cortex and in the medulla”⁸. We highlight that we do not, in this first description of IL11 effects in TECs, attempt to localise effects/expression to specific tubule segments.

For further assurances, we have performed additional Western blots and IHC on kidney samples from wildtype mice and mice with conditional deletion of *Il11ra1* in TECs (CKO mice) (Fig. 7B and Supplementary Fig. 10A). As expected IL11RA1 levels are reduced (but

not absent, as expressed in other cell types) at baseline in CKO mice. We also show IL11RA upregulation in WT mice kidneys following FA injury but not in CKO mice, inferring that IL11RA1 upregulation following kidney damage is primarily in the TECs or dependent on intact IL11 signalling in TECs. IHC shows IL11RA1 expression in tubules and glomeruli of WT mice but only in glomeruli in the CKO. This confirms both the efficacy and cell-type specificity of *Il11ra1* deletion in the CKO.

Rebuttal Figure 6. IL11RA expression in WT and *Cdh16*^{Cre/+}*Il11ra1*^{loxP/loxP} (*Il11ra* CKO) mice. (A) Western blots of IL11RA and GAPDH in kidneys of WT and *Il11ra* CKO mice subjected to either folic acid or NaHCO₃ vehicle control injection (representative dataset from n=5/group except for CKO-NaHCO₃: n=4). (B) Immunohistochemistry images of kidneys probed for IL11RA (with X209) in WT and CKO mice (scale bars: 100 μm; representative dataset from n=3/group).

6. It would strengthen the studies if the expression of IL-11 in the clinical specimens from CKD patients can be provided, such as blood, urine, or kidney tissue.

Author's response: Unfortunately, we do not have access to such samples. We note that IL11 is upregulated in the urine of patients with lupus nephritis¹⁰, in multiple mouse models of renal disease^{11,12} and in precision cut tissue slices of diseased human kidney¹³.

Other comments:

7. Figure 2C and Figure 6K only show the results of BUN, please add the results of serum creatinine.

Author's response: Fig. 2C and 6K are now **Fig. 3C and 9G**, respectively. Data for serum creatinine have been added as **Supplementary Fig. 3C** (for Fig. 3C) and **Fig. 9H** (for Fig. 9G).

8. Please add the quantitative results for Masson staining (Figures 2-3, Figures 5-6, and Supplementary Figure 1-4), and each group is at least 5 mice.

Author's response: In general, we prefer to show quantitative assessment of collagen using the HPA assay along with representative Masson's Trichrome staining, as in the original submission. However, at the request of the reviewer, we have now performed extensive additional quantification of MT staining in all experiments from n=5/group except for CKO-NaHCO₃ group which was carried out from n=4/group. These data, which mirror the HPA data, have been added to the main/supplementary figures where appropriate.

9. The Western blot results of animal experiments throughout the manuscript only show representative images (n=3). Please add the quantitative and statistical results of Western blots for all animals.

Author's response: We have performed semi-quantitative densitometry analyses of Western blots from n=4/group for *in vitro* TEC studies and n=5/group for all *in vivo* studies and added these data to the source datafile S2.

10. Figure 4A should add a group of IL11+X203.

Author's response: We do not think this control is needed as IL11 is neutralised by the antibody (as we have also shown in studies that we cite).

Reviewer #3 (Remarks to the Author):

IL11 is known to regulate cell fibrosis in many cell types and tissues (and there is some indicative data in kidney also from this group) also EMT in some cells, in particular cancer cells. This study therefore adds some information in the kidney. The study also indicates that IL11 may be targeted to regenerate kidneys after injury has occurred although the authors require further experiments to prove this. The mechanisms by which this occurs has not been thoroughly explored. It would be useful to determine changes in the transcriptome and more specifically it would be superior to investigate this at a single cell level rather than to investigate some factors associated with fibrosis, EMT and inflammation. The study is a great starting point but requires further investigation to determine the mechanisms by which IL11 acts specifically in the diseased/injured kidney.

Author's response: We thank the Reviewer for the supportive comments. While we are aware that scRNA-seq is now widely used, and we use it in our own studies where appropriate, we do not believe it would be informative in the current study. We specifically address the role of IL11 signalling in TEC EMT as a driving pathology for kidney dysfunction both *in vitro* and *in vivo* using genetic and pharmacologic gain- and loss-of-function approaches across models and experimental time courses. This said, during this revision we have performed new bulk RNA-seq experiments in primary cultures of TECs stimulated with IL11 over a time course to address a truly important point that this reviewer brought up, we refer the reviewer to the text below.

There are some specific points the authors may also consider.

Results:

There are too many abbreviations that make the paper difficult to follow – please reduce some such as FA and TECs.

Author's response: We thank the reviewer for bringing this to our attention. Given the length and complexity of this manuscript we feel that some abbreviations are unavoidable. However, we agree that the abstract was over-complicated and over-abbreviated and have amended this. We have also tried to limit the use of abbreviations in the revision and simplified the main text, sub-titles and the figure legends. We have also streamlined the discussion.

Figure 1:

Fig 1A. Need to complete statistical analysis to prove signal persists rather than just showing a representative IF image. Can't say something is less common unless you prove it. The samples size is small with N=3 and parametric tests will not be able to be completed as data normality will not be able to be assessed with n=3.

Author's response: We apologise for overinterpreting the image and have revised the text relating from “EGFP co-expression with E-Cadherin was less common and EGFP expression was mostly seen in interstitial regions lacking TEC markers” to “EGFP expression appeared to localise to interstitial regions lacking TEC markers”. We do not think it is helpful to try to quantify co-expression from a semi-quantitative IF image of damaged kidneys with distorted renal architecture and show this data as illustrative representations only.

Fig 1B. IF is notoriously difficult to quantitate and again this needs to be completed to make statements such as ‘very high levels’. As it stands you can say it localises to the cells. What concentrations of antibodies and IgG controls were used. I ask this as IL11Ra has been notoriously difficult to stain for using the commercial antibodies used in this study.

Author’s response: We agree that IF and IHC can be difficult to quantify and that any such analysis should be viewed as semi-quantitative. We have removed the phrase “very high levels” and similar such phraseology throughout. The antibody used in this study to stain for IL11RA was ab125015 from Abcam (1:200 dilution). These details have been added to the methods-immunofluorescence-primary human TEC section. We highlight that the human proteome atlas has successfully stained for IL11RA in kidney samples - notably in the renal tubules (see below), providing further reassurance to the reviewer.

Rebuttal Figure 7. IL11RA staining of human kidney from the human proteome atlas.

Left panel, antibody HPA036652; right panel, antibody CAB032830

(<https://www.proteinatlas.org/ENSG00000137070-IL11RA/tissue/Kidney>)

TGF beta is well known to stimulate the expression and secretion of IL11 in many cell types. Whether there is an autocrine or paracrine effect is not completely proven and it could equally be either.

Author’s response: TGF β 1, and other factors, induce a self-amplifying autocrine loop of IL11 activity in epithelial cells and also in stroma cells, such as fibroblasts. In **Fig. 5G-H** and **Supplementary Fig. 9**, we showed that media from TECs stimulated with TGF β 1 can cause renal fibroblast-to-myofibroblast transformation that is IL11-dependent (but TGF β 1 independent), thus demonstrating the paracrine effect from TECs to the stroma.

Fig 1D. Phosphorylation (p) often occurs within minutes if it is a direct effect after IL11 stimulation acting via its specific receptor, in particular in cells in culture that contain IL11Ra. The p of ERK is apparent only after 8 hours. This suggests that IL11 may signal via another pathway initially, perhaps the JAK/STAT pathway which it is well known to signal via, and primarily signal via in many cell types.

Author's response: The reviewer brings up a very important point and we apologise for our oversight on this matter. In recent studies, we have consistently shown that IL11 increases pERK across cell types and that this activation (from 15 mins to 24 hours) can be biphasic, as seen in vascular smooth muscle cells and some fibroblasts^{5,14,15} or can be sustained, as seen in pancreatic stellate cells¹⁶. For these reasons, we did not sufficiently scrutinise the time course of IL11 effects on TEC signalling to notice that ERK phosphorylation was not increased at early time points and only elevated later (>8h), after IL11 stimulation.

The reviewer's comment prompted us to go into much greater detail into the effects of IL11 on ERK vs STAT3 signalling and also to examine the transcriptional (RNA-seq) responses in IL11 stimulated TECs. These studies have generated new and unexpected insights and while these data extend further the length of the manuscript, we think they improve the study and now present them in a heavily revised main figure 1 and in a new results section that reads as follows:

IL11 induces pro-inflammatory and mesenchymal genes in tubular epithelial cells

The signaling data suggested that IL11-stimulated, pMEK-induced pERK levels might be suppressed by a phosphatase. To investigate this, and to study IL11-related transcriptional changes in TECs more generally, we performed RNA-sequencing of TECs stimulated with IL11 for 1, 6, or 24 hours.

Principal component analysis (PCA) revealed changes in gene expression across the time course (**Fig. 1E**). Examination of gene expression by molecular signatures database (MSigDB) analysis showed significant increases in a range of gene set hallmarks that included: TGF β signaling, IL6/JAK/STAT3 signaling, mTORC1 signaling, inflammatory and IFN γ responses, TNF α signaling, epithelial mesenchymal transition and G2M checkpoint (**Supplementary Fig. 1**). These data reinforce the importance of IL11 for TGF β effects¹⁵ and the activation of mTORC1 by IL11^{24,26} while suggesting a novel role for IL11 in TEC mesenchymal transition and inflammation.

We next examined changes in individual gene expression in an effort to identify a candidate phosphatase that might inactivate pERK (**Fig. 1D**). Remarkably, the most significantly upregulated transcript (30.8-fold, $P=2.2 \times 10^{-308}$) genome wide 1 hour post stimulation was *DUSP5* (**Fig. 1F**; **Supplementary Table 1**), which is an ERK phosphatase that limits inflammatory responses^{27,28}. *DUSP5* is not known to be regulated by STAT3 but our data suggested this, which we confirmed using an inhibitor of STAT3 (Stattic) that prevented IL11-induced *DUSP5* expression (**Fig. 1G**).

We then stimulated TECs with IL11 for 2 hours in the presence or absence of Stattic and assessed signaling pathways and *DUSP5* expression by western blotting. Inhibition of STAT3 activity prevented IL11-induced *DUSP5* expression, which was associated with increased pERK expression, thus establishing the presence of an early onset STAT3-*DUSP5*-pERK feedback loop in TECs, which is not seen in other cells (**Fig. 1H**).

IL11 is pro-inflammatory in fibroblasts where it causes STAT3-dependent IL33 expression²⁵. Further inspection of the RNA-seq data revealed that *IL33* is also highly induced (14.1-fold, $P=1.7 \times 10^{-54}$) in TECs at 6 hours post stimulation, along with *CCL20* (25.7-fold, $P=9.9 \times 10^{-06}$) and *CXCL8* (16.5-fold, $P=3.2 \times 10^{-55}$) (**Supplementary Table 1**), which are also increased in IL11-stimulated fibroblasts²⁵. Some of the most downregulated genes following IL11 stimulation were TEC-specific channel genes (e.g. renal tubular urea transporter (*SLC14A2*) and aquaporin 6 (*AQP6*)) (**Supplementary Table 1**).

Figure 1. Autocrine IL11 activity initiates a pro-inflammatory and mesenchymal program in tubular epithelial cells via activation of both STAT3 and ERK. (A) Representative immunofluorescence (IF) images (scale bars: 100 μ m) of EGFP and E-Cadherin expression in the kidneys of *Il11*^{EGFP/+} mice following folic acid (FA) injection (kidneys were collected at day3, day 21, and day28, representative dataset from n=3/group). (B) Representative IF images of IL11RA staining of primary human renal TECs (scale bars: 100 μ m; representative dataset from n=3/group). (C) ELISA of IL11 secretion from TGFβ1-stimulated TECs (24 hours, n=4/group). (D) Western blots of STAT3, MEK, ERK, p90RSK, and GSK3β activation and expression levels of SNAI1 and E-Cadherin in IL11-stimulated TECs over a time course (representative dataset from n=4/group). (E-F) Data for RNA sequencing (RNA-seq) experiments on TEC stimulated with IL 11 for 0 (Bsl), 1, 6, and 24 hours. (E) Principal component analysis (PCA) of RNA-seq across the time-series. PC1 and PC2 account for 59% and 18% variance of gene expression, respectively. (F) Volcano plot displaying the adjusted p-value (-log₁₀(p-adj)) and fold change (log₂(Fold change)) of genes between IL11 stimulated cells versus baseline at 1-hour time point. Dashed lines are drawn to show log₂(Fold change) value of -1 and +1 (vertical) and p-adj of 0.05 (horizontal). Upregulated, downregulated, and non-differentially expressed genes are labeled in orange, purple, and gray, respectively. DUSP5 is annotated in red for clarity. (G-H) (G) Relative *DUSP5* mRNA expression normalized

to *GAPDH* (n=4/group) and (H) Western blots showing STAT3, MEK, and ERK activation and DUSP5 expression (n=4/group) in IL11-stimulated TECs (2 hours) in the presence of either DMSO (vehicle) or STAT3 inhibitor (Stattic). (A-H) IL11 (5 ng/ml), TGFβ1 (5 ng/ml), Stattic (2.5 μM). (C) Data are shown as box-and-whisker with median (middle line), 25th–75th percentiles (box), and minimum-maximum values (whiskers); 2-tailed Student's *t*-test. (G) Data are shown as mean±SD; one-way ANOVA with Tukey's correction. Bsl: Baseline; Rel. Exp: Relative expression. Source data are provided as a Source Data file.

There is also no quantitation of the Westerns. Please show the densitometry and statistical analysis. I just noted you have an N=1. Is this correct? Please repeat to get the required N's as determined by power analysis to ensure statistical significance can be determined.

Author's response: We have now performed immunoblotting and semi-quantitative densitometry analyses of the respective blots from n=4/group for *in vitro* TEC studies and n=5/group for all *in vivo* studies and added these data to the source datafile 1.

Fig 1E. The authors used one dose of FA 200mg/kg and then samples taken at Day 28. Why was this dose chosen and why were they left for 28 days before samples were collected? Is this a standard protocol? This is a very long time to determine whether the effects noted 28 days later are caused by loss of IL11 signaling directly or were associated with IL11 signalling indirectly. Is it also that the effect could be as a cause of the disease that is presumably apparent after 28 days.

Author's response: We conducted preliminary analyses in the strains used in our studies to obtain the optimal dose of FA from a dose range of 150-250mg/kg for inducing severe kidney damage with acceptable mortality. Based on these studies, folic acid (FA) was administered at 150 mg/kg to the IL11-EGFP strain and 200 mg/kg for the C5JBL6/J strains (therapeutic studies).

To outline part of our model validation data: kidney samples were collected from mice from day 3 (D3) to D56 following FA dosing for western blotting and collagen quantification (**Rebuttal Fig. 9**). As can be seen, collagen increased from D10 to D21 post FA and then plateaued thereon (**Rebuttal Fig. 9A**), as expected in a model of acute kidney injury that transitions to chronic kidney disease. IL11 levels were also upregulated from D3 to D21 post FA and then plateaued and remained elevated up to 15 weeks later (D105) (**Rebuttal Fig. 9B**). These data formed the basis for our AKI prevention and treatment experiments using FA (pre-injury and D3 postinjury, respectively) and our CKD reversal experiments (from D21

post injury) and informed timepoints of sample harvesting.

A

B

C

Rebuttal Figure 9. Development and validation of folic acid-induced acute kidney injury and chronic kidney disease. (A) Development of fibrosis (total collagen content by hydroxyproline assay) in the kidney post folic acid (FA; 200mg/kg) injury shows stable fibrosis from day 14 onwards with no evidence of spontaneous regression (n=3-16/group). (B) Western blots (representative of dataset n=5/group) and (C) densitometry analysis of IL11 normalized to GAPDH expression in kidneys collected at D3, D21, D28, and D105 post FA. (C) Data are shown as mean±SD, one-way ANOVA with Dunnett's comparisons.

Fig 1F, please show densitometric analysis and statistical tests – likely non-parametric tests as you cannot test for normality with an N=3.

Author's response: We have now performed semi-quantitative densitometry analyses and statistical tests from n=4/group for *in vitro* TEC studies and n=5/group for all *in vivo* studies and added these data to the source datafile S2.

Fig 1G. Not sure what G is showing precisely (from the legend). Please clarify.

Author's response: Fig. 1G is now **Fig. 2C**. The image shows the gross anatomy of the kidneys from both genotypes at 28 day post FA i.e kidney from wild-type mice is visibly smaller in size due to the lost kidney mass from the tubular injury and has a rough surface (fibrotic) whereas the kidney from *Il11* KO mice looks bigger with smoother surface (healthier and less fibrotic).

Not sure what this refers to:

(C-D) TGFβ1/IL11 (5 ng/ml); 24 hours.

Author's response: This refers to the concentration of TGFβ1 or IL11 and the duration of the stimulation that were used for experiments shown in Fig. 1C and D.

Figure 2. The authors need to demonstrate / comment on the half-life of the antibodies and determine if they reach the site of interest by demonstrating blocking of specific downstream signalling, such as pSTAT3 or ERK, after perhaps a single injection and collection of tissue of interest ie. Kidney over a short time frame. Could the effects could be off target effects? There are no good effective IL11 signalling blocking antibodies as far as I can determine from the literature. This is again the same for Figs 3 and 4.

Author's response: The neutralising IL11 antibody used in the studies here has been characterised in great detail both *in vitro* and *in vivo* and we refer the reviewer to some of the relevant cited publications²⁻⁵. The X203 clone has an equilibrium dissociation constant of 2.4 nM for mouse IL11, and based on blood pharmacokinetics of [125I] labelled antibody injected to C57/BL6J male mice, a half life of approximately 9 days², and is readily detected in the kidney over this time course.

There are effective IL11 neutralising clones that are available commercially such as MAB218 (RnD Systems), that have been used by others to effectively block IL11 activity *in vivo*. Indeed, we used MAB218 in the current study as an orthogonal/complementary approach and showed its dose-dependent effects that mirror those of X203. We also used an anti-IL11RA antibody in the *in vitro* studies (current **Supplementary Fig. 7C**), which mirrored the effects of X203.

As shown above, IL11 levels are elevated throughout the time course of the experiments and therefore inhibition of IL11 with the neutralising antibody (or genetically) would be expected to block downstream signalling (e.g. the IL11/ERK/p90RSK/GSK3β/SNAI1 axis) at the time points we assessed. Indeed, this is what we show across multiple experiments that we have summarised for the Reviewer's convenience in Appendix A (below).

We highlight that STAT3 activation (pSTAT3) by IL11 in TECs is not sustained (**Fig. 1D**) and we demonstrated the specific importance of ERK signalling for TEC mesenchymal transition using U0126 (**Supplementary Fig. 7D-F**).

The antibody used here, X203, has been humanised and is in development for human safety trials and the rigours ascribed to its specificity cannot be overstated. In the academic setting, the antibody was shown to replicate the effects of anti-IL11RA or genetic deletion of either *Il11* or *Il11ral* showing its effects to be specific to the inhibition of IL11 signalling. Further specificity is ascribed by the lack of binding of this antibody to any other proteins in *Il11* KO mice (current **Fig. 2B**; **Supplementary Fig. 2B**). Dose response effects, further validating on-target, non-threshold effects of X203 were also presented in the manuscript (current **Fig. 3A-C**; **Supplementary Fig. 3**). As mentioned, we also showed data for a commercial antibody, MAB218 that binds a distinct IL11 epitope⁵. MAB218, while less effective than X203, has similar dose-dependent effects on fibrosis, renal function and cell signalling (**Supplementary Fig. 4**), further confirming on-target and specific effects.

I am not quite sure how the neutralisation of TGF beta experiment adds to the study aims. Did the authors hope to determine the specific mechanistic differences between the two agents in the kidney. The authors need to show that the neutralising antibodies neutralise their specific actions in the kidney, or are they effects secondary/indirect.

Author's response: We think this is a very important aspect of our manuscript. Sadly, and indeed tragically, anti-TGFβ failed in clinical trials in patients with CKD²⁰. We suggest that the failure of anti-TGFβ in the clinic reflects on-target toxicity, which was suspected by the trialists. Our experiments demonstrate this to be the case in the mouse: while anti-TGFβ reduces fibrosis, as expected, it increases kidney inflammation (*Il6*, *Ccl2*, *Tnfa*, *Il1b*) and tubular damage (*Ngal*, *Kim1*). We show that anti-IL11 has a better safety profile: it reduces fibrosis similar to anti-TGFβ, as expected, but has the additional benefits of preventing inflammation and diminishing tubular damage. We think these insights are most pertinent as they may reopen discussions on inhibiting the TGFβ pathway more generally in CKD and possibly pave the way for new clinical trials that avoid anti-TGFβ on-target toxicities.

We used the widely studied pan anti-TGFβ (anti-TGFβ-1,2,3; clone 1D11) at a published dose and can confirm that this dose inhibits pSMAD2 in the damaged kidney, as expected (**Rebuttal Fig. 10 and revised Fig. 3L**). Noticeably, anti-IL11 also reduces pSMAD2 in the kidney showing a feed forward effect of IL11 on TGFβ signalling, as we have observed before in lung².

Rebuttal Figure 10. (A) Western blots of phosphorylated (p-) and total SMAD2 expression in FA injured kidneys treated with IgG, anti-IL11 (X203) and anti-TGF β (1D11) (all 20 mg/kg) compared to NaHCO₃ vehicle controls (n=5/group). (B) Densitometric analysis of SMAD2 activation in FA-injured kidneys. Data are shown as mean \pm SD, one-way ANOVA with Tukey's comparisons.

These *in vivo* data are complemented by our studies of the inter-relationship of TGF β 1 and IL11 in TECs and the data showing that IL11-induced TEC activation in paracrine is dominant over TGF β 1.

The same quantitation if the Westerns is also required.

Author's response: Western blots have been quantified throughout and shown in source datafile S2 and values reported in source datafile S1.

I am not sure how the statistical analysis was completed for the immunofluorescence as in Suppl Figure 6F. There appears to be substantial overlap for Collagen 1 for IL11 DMSO and IL11- and there is a statistical difference between the two. The representative images in Suppl Fig 6E also appear to be very similar. It would be ideal to see the details for the quantitation and statistical analysis. Again, same with the Western blots.

Author's response: Supplementary Fig. 6F is now **Supplementary Fig. 7F**. These collagen 1 immunofluorescence data were analysed by one-way ANOVA with Tukey's correction. There was a very small size effect detected by the statistical analysis ($p=0.03$) between IL11 (mean I/A: 447) and IL11+DMSO (mean I/A: 464) from the earlier experiment due to the number of points/values (n=14/group) used for analysis. We agree with the Reviewer that this is odd as addition of 0.1% DMSO should not have any effect (as shown in **Figs. 7D, 7E, 7F** (SMA and SNAI readout), and **7G**. We have now repeated this experiment and confirmed

that there is no significant difference in the Collagen I fluorescence I/A between IL11 and IL 11+DMSO group. Quantification (densitometry analysis) for the Western blots data have been performed and presented in Source Data file 2, with the values reported in Source Data file 1.

Figure 4G (and Suppl Fig 8) – error in ID10 – Should it read ID11?

Author’s response: This has been amended

Figure 6 – Time frame for treatment with anti-IL11 is 24 hours. It is difficult to tell whether the effects seen are due specifically by IL11 or secondary effects. Have the authors trialled treatment with anti-IL11 for shorter time periods and completed a time course in vivo and investigated the effects on the JAK/STAT pathway (specifically activation) which IL11 primarily signals via. This may indicate direct and/or indirect effects of IL11 and also determine whether the neutralising antibody reaches the site of interest and how long it is present at the site of interest to work.

Author’s response: As regards anti-IL11 treatment for 24 h in original Figure 6, we surmise the reviewer refers to original Fig. 6A to D (which are now **Fig. 8A-D** in the revised manuscript) where we showed that addition of anti-IL11 (X203) to cultures pre-stimulated with TGFβ1 reversed mesenchymal phenotypes in the continued presence of TGFβ1. This shows that TGFβ1-induced TEC pEMT is not only IL11 dependent but can be *reversed* by anti-IL11.

With regard to the comment on IL11 and STAT3, we refer the reviewer to the new data that we show on IL11-induced signalling in TECs (**Fig. 1D, G-H**), the data showing the primacy of ERK signalling for EMT phenotypes using U0126 and the published literature that we cite.

The discussion has some repetition and is confusing to read and often repeats the results. Can it be re-written/organised to start with for instance - the main findings of the study rather than what is already known about kidney injury and TECs which should be in the introduction. The specific results do not need to be repeated in the discussion section.

Author’s response: The discussion has been restructured and simplified.

References

1. Widjaja, A. A. *et al.* IL11 stimulates ERK/P90RSK to inhibit LKB1/AMPK and activate mTOR initiating a mesenchymal program in stromal, epithelial, and cancer cells. *iScience* vol. 25 104806 (2022).
2. Ng, B. *et al.* Interleukin-11 is a therapeutic target in idiopathic pulmonary fibrosis. *Sci. Transl. Med.* **11**, (2019).
3. Widjaja, A. A. *et al.* Inhibiting Interleukin 11 Signaling Reduces Hepatocyte Death and Liver Fibrosis, Inflammation, and Steatosis in Mouse Models of Non-Alcoholic Steatohepatitis. *Gastroenterology* (2019) doi:10.1053/j.gastro.2019.05.002.
4. Ng, B. *et al.* Similarities and differences between IL11 and IL11RA1 knockout mice for lung fibro-inflammation, fertility and craniosynostosis. *Sci. Rep.* **11**, 14088 (2021).
5. Widjaja, A. A. *et al.* Molecular Dissection of Pro-Fibrotic IL11 Signaling in Cardiac and Pulmonary Fibroblasts. *Frontiers in Molecular Biosciences* **8**, 926 (2021).
6. Zhou, B. P. *et al.* Dual regulation of Snail by GSK-3beta-mediated phosphorylation in control of epithelial-mesenchymal transition. *Nat. Cell Biol.* **6**, 931–940 (2004).
7. Sekiya, S. & Suzuki, A. Glycogen synthase kinase 3 β -dependent Snail degradation directs hepatocyte proliferation in normal liver regeneration. *Proc. Natl. Acad. Sci. U. S. A.* **108**, 11175–11180 (2011).
8. Grande, M. T. *et al.* Snail1-induced partial epithelial-to-mesenchymal transition drives renal fibrosis in mice and can be targeted to reverse established disease. *Nat. Med.* **21**, 989–997 (2015).
9. Lovisa, S. *et al.* Epithelial-to-mesenchymal transition induces cell cycle arrest and parenchymal damage in renal fibrosis. *Nat. Med.* **21**, 998–1009 (2015).
10. Chien, J.-W. *et al.* Daily urinary interleukin-11 excretion correlated with proteinuria in IgA nephropathy and lupus nephritis. *Pediatr. Nephrol.* **21**, 490–496 (2006).
11. Corden, B., Adami, E., Sweeney, M., Schafer, S. & Cook, S. A. IL-11 in cardiac and renal fibrosis: Late to the party but a central player. *Br. J. Pharmacol.* **177**, 1695–1708 (2020).
12. Widjaja, A. A. *et al.* A Neutralizing IL-11 Antibody Improves Renal Function and Increases

- Lifespan in a Mouse Model of Alport Syndrome. *J. Am. Soc. Nephrol.* **33**, 718–730 (2022).
13. Bigaeva, E. *et al.* Transcriptomic characterization of culture-associated changes in murine and human precision-cut tissue slices. *Arch. Toxicol.* **93**, 3549–3583 (2019).
 14. Lim, W.-W. *et al.* Interleukin-11 is important for vascular smooth muscle phenotypic switching and aortic inflammation, fibrosis and remodeling in mouse models. *Sci. Rep.* **10**, 17853 (2020).
 15. Widjaja, A. A. *et al.* IL11 Stimulates IL33 Expression and Proinflammatory Fibroblast Activation across Tissues. *Int. J. Mol. Sci.* **23**, 8900 (2022).
 16. Ng, B. *et al.* IL11 Activates Pancreatic Stellate Cells and Causes Pancreatic Inflammation, Fibrosis and Atrophy in a Mouse Model of Pancreatitis. *Int. J. Mol. Sci.* **23**, (2022).
 17. Schafer, S. *et al.* IL-11 is a crucial determinant of cardiovascular fibrosis. *Nature* **552**, 110–115 (2017).
 18. Caunt, C. J. & Keyse, S. M. Dual-specificity MAP kinase phosphatases (MKPs): shaping the outcome of MAP kinase signalling. *FEBS J.* **280**, 489–504 (2013).
 19. Habibian, J. S. *et al.* DUSP5 functions as a feedback regulator of TNF α -induced ERK1/2 dephosphorylation and inflammatory gene expression in adipocytes. *Sci. Rep.* **7**, 12879 (2017).
 20. Voelker, J. *et al.* Anti-TGF- β 1 Antibody Therapy in Patients with Diabetic Nephropathy. *J. Am. Soc. Nephrol.* **28**, 953–962 (2017).

Appendix A (for Reviewer 3)

Experiments where inhibition of IL11 is shown to inhibit IL11-related signaling

- **Fig. 2B** (*Il11* KO, signalling at D28 post FA).
- **Fig. 3L** (X203 prevention dosing; signalling at D28 post FA).
- **Fig. 4D** (X203 therapeutic dosing starting from D3 post FA; signalling at D28 post FA).
- **Fig. 5D** (X203 therapeutic dosing starting from D3 post UUO; signalling at D10 post-UUO).
- **Fig. 6B, D** (pro-fibrotic factors+X203 in primary human TEC; signalling at 24 hours post stimulation).
- **Fig. 7B** (TEC-specific *Il11ra* KO (*Il11ra* CKO), signalling at D28 post FA).
- **Fig. 8A** (TGF β 1+X203 reversal study in primary human TEC; signalling at 24 hours post X203 addition).
- **Fig. 9F** (X203 reversal dosing; signalling at 15 weeks post FA).
- And also in **Supplementary Figs. 4G and 7C**.

Reviewers' Comments:

Reviewer #1:

Remarks to the Author:

The authors have responded well to the Reviewers' concerns with additional analyses, experiments and clarifications. I remain somewhat concerned that the in vitro TEC data is at times not convincingly verified in vivo especially with respect to signaling mechanisms. This is admittedly difficult to do. Along these lines i had one concern about the conclusions drawn regarding DUSP5.

The authors have performed RNASeq on IL11-treated cultured TECs for different periods of time. DUSP5 was the most upregulated gene and the authors attribute a great deal of importance to this stating in the Abstract that this reflects a IL11-induced.. "STAT3-DUSP5-ERK cross talk, pro-inflammatory gene expression as well as ERK/p90RSK/GSK3b phosphorylation and SNAI1-driven growth arrest." Since a good deal of mechanistic information is inferred from this observation it is important to know that DUSP5 is upregulated in the appropriate epithelial cells in vivo in the mouse model of FA and not upregulated in the IL11-deleted mice.

Also can DUSP5 levels be correlated with the CKD reversal seen in figure 9? Unless I may have missed it, I do think it is important to address these questions given the importance the authors place on this signaling molecule.

Reviewer #2:

Remarks to the Author:

The authors have addressed most of my concerns, while I still have some reservation about the novelty of the study, as previous studies have shown the role of IL-11 in CKD and fibrotic diseases of other organs.

Reviewer #3:

Remarks to the Author:

The authors have addressed most of the reviewer's comments and have undertaken additional experiments. Addition of single cell RNA sequencing would have ideally teased out differences in effects of IL11 in the epithelial and stromal compartment. The authors have completed builds RNA sequencing that adds some additional data. Overall the study provides some new insight of IL11 in the disease.

Point-by-point responses to the comments made by Reviewer 1 at Nature Communications

Reviewer #1 (Remarks to the Author):

The authors have responded well to the Reviewers' concerns with additional analyses, experiments and clarifications.

Response: We thank the reviewer for recognising our efforts to address his/her comments that have greatly improved the manuscript.

I remain somewhat concerned that the in vitro TEC data is at times not convincingly verified in vivo especially with respect to signaling mechanisms. This is admittedly difficult to do.

Response: We have gone to some lengths to show the relevance of our in vitro signalling findings in TECs for pathological signalling in vivo in models of kidney disease. To this end, we show activation of the ERK/p90RSK/GSK3 β /SNAI1 axis in IL11-stimulated TECs *in vitro* and corroborate the *in vivo* relevance of this observation in a number of experiments in mice: (1) global *Il11* KO, (2 and 3) anti-IL11 (X203) in prevention and treatment modes in mice with folic acid (FA) injury, (4) anti-IL11 (MAB218) in FA injury, (5) X203 in treatment mode in mice with UUO, and (6) X203 in CKD reversal experiments. Importantly, we assign these signalling events specifically to TECs in another *in vivo* experiment: (7) FA-injured kidneys in mice with TEC-specific deletion of *Il11ra1*. The relevance of inhibition of these pathways for TECs *in vivo* is now also shown using EdU staining of replicating TECs in the CKD reversal experiments.

Along these lines i had one concern about the conclusions drawn regarding DUSP5. The authors have performed RNASeq on IL11-treated cultured TECs for different periods of time. DUSP5 was the most upregulated gene and the authors attribute a great deal of importance to this stating in the Abstract that this reflects a IL11-induced.. "STAT3-DUSP5-ERK cross talk, pro-inflammatory gene expression as well as ERK/p90RSK/GSK3b phosphorylation and SNAI1-driven growth arrest."

Response: We believe there is a misunderstanding on this point and clarify here that we did not mean to "attribute a great deal of importance" to the DUSP5 data and apologise if it seemed that way. Indeed, we think this finding is a little tangential to the larger study and the overall manuscript message. DUSP5 was identified following unbiased genomic screens of IL11 stimulated TECs that were performed during revision to address a specific point raised by Reviewer 3 as to why phosphorylation of ERK was not seen at early time points following IL11 stimulation. We show that this *in vitro* signalling phenomenon is due to the rapid and transient STAT3-mediated upregulation of DUSP5 in IL11 treated TECs. This is the only reference and relevance of the DUSP5 observation in the manuscript. To try to address the Reviewer's concern, we have removed any mention of DUSP5 from the abstract and mention it only in the results, in passing.

Since a good deal of mechanistic information is inferred from this observation it is important to know that DUSP5 is upregulated in the appropriate epithelial cells in vivo in the mouse model of FA and not upregulated in the IL11-deleted mice.

Response: We do not assign 'a good deal of mechanistic information' to DUSP5. We show only that short-term upregulation of DUSP5 in IL11 stimulated TECs suppresses ERK phosphorylation, addressing Reviewer 3's question. In the mouse models, where IL11 is persistently upregulated, we show across the 7 *in vivo* experiments above that ERK is consistently phosphorylated. While chronic IL11 activity is not expected to upregulate DUSP5 (as this is a transient event), we believe assessing

DUSP5 levels is a mute point as the aggregate effect of all ERK phosphatase activity (including DUSP5) demonstrably does not prevent ERK phosphorylation, which is elevated in the injured kidneys (Figs 2B, 4D, 5D, 7B, 9F).

Also can DUSP5 levels be correlated with the CKD reversal seen in figure 9? Unless I may have missed it, I do think it is important to address these questions given the importance the authors place on this signaling molecule.

Response: As emphasised above, we do not place importance on DUSP5 other than to associate it with the *in vitro* signalling phenomenon identified by Reviewer 3. For the reasons outlined above, we do not think assessing DUSP5 in the CKD is meaningful as ERK is chronically phosphorylated (Fig 9F), thus aggregate ERK phosphatase activity (including DUSP5) is subservient to MEK activity.